# ROUTING WITH RICH TEXT QUERIES VIA NEXT-VERTEX PREDICTION MODELS

## ABSTRACT

Autoregressive modeling of text via transformers has led to recent breakthroughs in language. In this work, we study the effectiveness of this framework for routing problems on graphs. In particular, we aim to develop a learning based routing system that can process rich natural language based queries indicating various desired criteria and produce near optimal routes from the source to the destination. Furthermore, the system should be able to generalize to new geographies not seen during training time.

Solving the above problem via combinatorial approaches is challenging since one has to learn specific cost functions over the edges of the graphs for each possible type of query. We instead investigate the efficacy of autoregressive modeling for routing. We propose a multimodal architecture that jointly encodes text and graph data and present a simple way of training the architecture via *next token prediction*. In particular, given a text query and a prefix of a ground truth path, we train the network to predict the next vertex on the path. While a priori this approach may seem suboptimal due to the local nature of the predictions made, we show that when done at scale, this yields near optimal performance.

We demonstrate the effectiveness of our approach via extensive experiments on synthetic graphs as well as graphs from the OpenStreetMap repository. We also present recommendations for the training techniques, architecture choices and the inference algorithms needed to get the desired performance for such problems.

## 1 INTRODUCTION

Scaling transformer architectures along with training data continues to demonstrate improved performance and emergent abilities in domains such as text, images and video (Chowdhery et al., 2022; Brown et al., 2020; Chen et al., 2022). A key ingredient in these breakthroughs is the paradigm of self-supervised training via *next token prediction*. This leads to an elegant and highly distributed strategy for pre-training. Another key advantage is the uniform manner in which multi-modal data can be ingested via appropriate tokenization strategies.

In this work, we investigate the effectiveness of the above framework for a unique and challenging multimodal setting namely, routing on graphs via natural language queries. Routing is a classical problem with a long history of the study of combinatorial and search algorithms under various constraints (Tarjan, 1972; Geisberger et al., 2008; Cormen et al., 2022). We study the problem of routing in the context of road networks, where it is often the case that users need to be routed from a source to the destination under a variety of constraints. For instance, users may want to avoid highways, prefer a safe route over the shortest route, or run an errand on the way to the destination.

Traditionally, there are two approaches to handling such constrained routing problems. The first is the use of *cost modifiers* which amounts to producing edge costs on the graph for each type of constraint/customization, followed by running a shortest path primitive such as the Dijkstra's algorithm to produce the relevant route. However, such an approach requires one to design (usually via domain knowledge) many different types of cost functions which does not scale for complex queries, especially if they are expressed in natural language. Furthermore, not all types of constraints can be effectively modeled as performing Dijkstra's algorithm on an appropriate edge based cost function. The second approach is to perform an unguided search via graph search algorithms or spectral algorithms (Dechter & Pearl, 1985; Sinop et al., 2021) to produce a diverse set of alternates

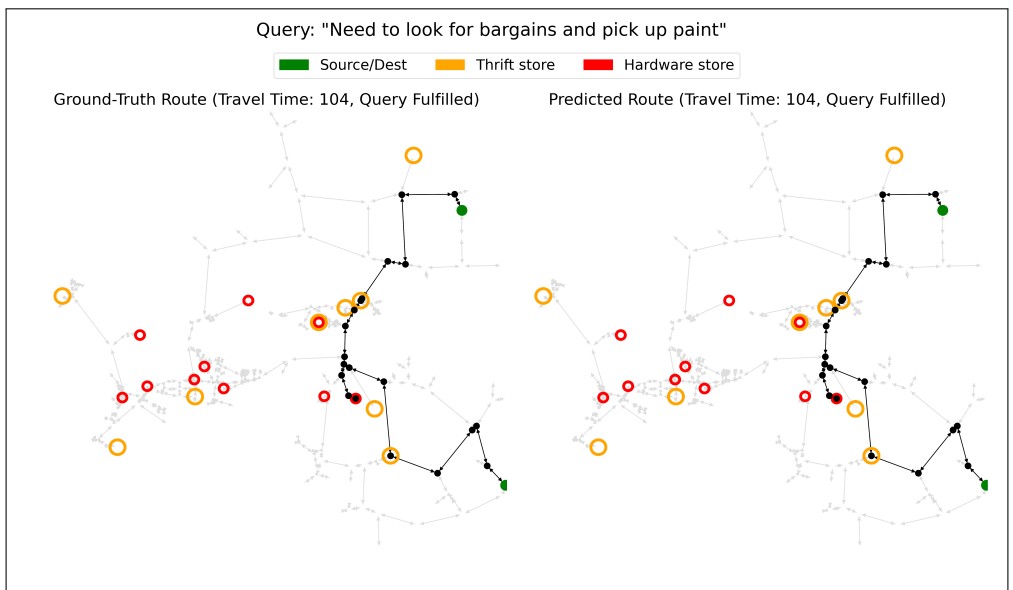

Figure 1: Example a of natural language based routing query on the OpenStreetMap (Open-StreetMap contributors, 2017) graph (United States). The source and destination are highlighted in green. The various points-of-interest are highlighted in red, blue and yellow. We show examples of routing queries expressed in natural language and the near optimal routes generated by our algorithm via direct inference on a transformer based model. Further examples can found in Appendix A.

between the source and the destination and to separately rank them according to the desired criteria. However, in general there is no guarantee that the optimal or a near optimal route will be part of the initial search phase unless a very large set of alternates is explored.

The emergence of transformers and large language models (LLMs) presents an intriguing prospect of developing a routing system that can directly process complex queries in natural language and produce near optimal routes via implicit multi-objective routing. In this work we present a simple framework for building such routing systems. See Figure 1 for an example use case and the output produced by our routing model. There are two crucial aspects that need to be addressed to enable such use cases: a) the architecture choice for jointly modeling text and geo-spatial data, and b) the training paradigm and the choice of the loss functions.

Regarding the choice of the multimodal architecture, a naive approach would be to take inspiration from domains such as images (Dosovitskiy et al., 2020) and, given a (*text-query*, *graph*, *source*, *destination*) tuple, aim to tokenize the graph via appropriately defining "patches". However, it is not clear what the right level of granularity should be. Too coarse of a patch may not provide any relevant information for the given (*source*, *destination*) query, and too fine-grained patches will significantly blow up the computational burden for massive graphs as a result of very long sequences. Similarly, regarding the design of the loss function, one could consider approaches such as reinforcement learning where the underlying reward models would capture the goodness of a particular route. However, this approach will blow up the data and training time requirements significantly and sacrifice the elegance of the next token prediction framework that has proved to be so successful in the domains of language modeling and computer vision.

We instead present a simple and scalable architecture that can easily generalize to massive graphs and can be trained following the standard *next token prediction* recipe. In particular, given a text query $q$, source $s$, destination $t$ and a prefix of a route from $s$ to $t$ we train the model by the autoregressive loss of predicting the next vertex on the route. Furthermore, we only define "tokens" based on local neighborhoods of any vertex. While this may seem lossy at first since the model only has a local view of the graph, we show that when done at scale this simple framework can yield surprisingly powerful results for routing queries. Furthermore, our proposed framework can be easily

scaled to train on massive graphs without the need for distributing a graph over many machines. We demonstrate the effectiveness of our approach on both synthetic and real world data.

The rest of the paper is structured as follows. In Section 1.1 we survey related work. In Section 2 we study the simplified problem of custom routing on a fixed graph topology. We use this setting to define our architecture and the pre-training methodology. Finally, in Section 3 we extend and introduce the notion of *road embeddings* that help us generalize to unseen graph topologies.

## 1.1 RELATED WORK

Routing is traditionally approached via combinatorial methods. In the context of road networks these methods fall into two broad categories. The first category concerns using classical algorithms such as Dijkstra's method (Dijkstra, 2022) and ways to make it more efficient via priority queue based implementations (Goldberg, 2001; Meyer, 2001), bidirectional search (Sint & de Champeaux, 1977), A\* search (Hart et al., 1968) and so on. The second concerns the use of hierarchical approaches such as contraction hierarchies (Bast et al., 2016; Delling et al., 2009; Bauer et al., 2010) that specifically make use of the near planar structure of the road networks. Incorporating custom query constraints into these algorithms is challenging. This requires one to either solve a new combinatorial problem (often NP-complete) or to design *cost modifiers* on edges such that the original algorithms when run using the new costs will result in near optimal solutions. For instance, one may make the cost of highway edges to be infinite if the request is to route while avoiding highways. However, this in turn requires one to design good cost modifiers for complex natural language based queries, a challenging problem in itself. Furthermore, not all types of queries can be translated into edge-based cost modifiers while maintaining optimality of the ground truth solution.

An alternate approach consists of running unguided search algorithms such as A\* search or spectral methods based on electrical flows (Sinop et al., 2021) to compute many candidate routes from source to destination. However, this approach needs one to have a good way of scoring the routes and for complex queries one may need to produce many candidate solutions in the initial set, thereby making the overall methodology inefficient.

In recent years, deep-learning-based approaches have been explored for combinatorial problems such as routing. The works of Xu et al. (2019; 2020) show that graph neural networks (GNNs) are an efficient architecture for representing dynamic-programming-based algorithms and hence can, in principle, be used for shortest path problems. The works of Veličković et al. (2019); Ibarz et al. (2022) create GNN based neural learners to simulate various combinatorial algorithms via "hinting" them with intermediate computations of the algorithms. There have also been works to use custom models such as pointer networks and attention-based GNNs to address related problems such as the travelling salesman problem (Kool et al., 2018; Vinyals et al., 2015; Khalil et al., 2017; Nowak et al., 2017; Deudon et al., 2018). However, these approaches have been empirically tested only at small scales (around 100 nodes). To scale them to massive graphs such as the OpenStreetMap repository (OpenStreetMap contributors, 2017), one would at the very least need a distributed GNN implementation which presents its own set of infrastructure difficulties. Our approach on the other hand easily scales to millions of nodes without any need for distributed training. One exception to the small scale studies in previous works is the result of Graves et al. (2016), who conducted routing experiments on the London underground system. However, this required a specialized model of differential neural computer with access to read/write memory. We instead simply rely on the elegant framework of transformer models and next token prediction without any complex customizations.

## 2 CUSTOMIZED ROUTING VIA NEXT-VERTEX PREDICTION

We propose to approach route generation via autoregressive modeling. By treating vertices and edges as "tokens" and routes as "sentences", we unroll a proposed route by training a model to predict the vertex that follows the current incomplete route: given a text query and a pair of source and destination vertices, we first predict the second vertex to come in the route, and then a third vertex to follow the source and second vertex, and so on.

**Customized route tasks.** Given a road network graph $G = (V, E)$, a source vertex $u \in V$, a destination vertex $v \in V$, and a natural language query $s \in \Sigma^*$ where $\Sigma$ is the alphabet, the cus-

tomized routing task is to find a route from the source to destination that accommodates the requests conveyed in $s$. Crucially, no assumptions are made about the syntax (i.e., phrasing) or semantics (i.e., meaning) of the query. For example, a query may ask for a route that avoids bike lanes and passes by a coffee shop, or a route that takes a longer to drive but passes by scenic waterfalls.

Formally, the goal of customized routing is to output a route $(v_1, \ldots, v_n)$, where $v_i \in V$, $v_1 = u$ and $v_n = v$, such that some objective $f(s, u, v, (v_1, \ldots, v_n))$ is maximized, where $f : \Sigma^* \times V^2 \times V^* \to \mathbb{R}$ returns a real valued score given a customized routing task and a candidate route. Importantly, this objective $f$ can be an arbitrary function of the entire route and need not decompose additively into components on the route's edges, i.e., there may not be a function $f' : \Sigma^* \times V^2 \times V^* \to \mathbb{R}$ such that $f(s, u, v, (v_1, \ldots, v_n)) = \sum_{i=1}^{n-1} f'(s, u, v, (v_i, v_{i+1}))$. For example, the query "pass by a coffee shop on route" cannot be expressed in such a form. This problem therefore cannot be reduced to that of finding shortest-paths on an appropriately edge-weighted graph, thus motivating our next-vertex prediction framework.

**Next-vertex prediction models.** We decompose a customized routing task with respect to a tuple $(s, u, v)$ into a series of next-vertex prediction problems, each defined by $(s, u, v)$ and an incomplete route $(v_1, \ldots, v_i)$, which is the prefix of some ground-truth route $(v_1, \ldots, v_n)$, and whose goal is to predict the next vertex $v_{i+1}$ on the route. Given a next-vertex prediction model $M$, we can "unroll" a candidate route $(v_1, \ldots, v_n)$ for a customized routing task problem $(s, u, v)$ by defining $v_1 = u$ and $v_{i+1} = M((s, u, v), (v_1, \ldots, v_i))$, ending when $M$ returns a termination signal.

In this work, we learn such models $M$ by training decoder only transformer models on datasets consisting of examples of customized routing tasks and their ground-truth routes, decomposing each such ground-truth route $(v_1, \ldots, v_n)$ into $n - 1$ next-vertex prediction problems $(v_1) \to v_2$, $(v_1, v_2) \to v_3$, etc. This approach is analogous to the training of autoregressive language models on text corpora (Brown et al., 2020; Chowdhery et al., 2022).

## 2.1 Experiments on a Simulated Grid Road Network

We first apply our next-vertex prediction approach to a dataset of customized routing tasks on a simulated grid road network consisting of $625$ vertices and $2462$ edges. The routing tasks represented in these datasets concern user queries that specify "points-of-interest" (POI), such as users asking to stop by a coffee shop and bakery on route, or users asking to take a detour to either a dog park or lake. In particular, throughout this section, the graph will remain fixed an we only consider generalizing to unseen queries at test time. This simplified setting will help us describe the key aspects of our architecture and training details. In the next section we then handle the more realistic and challenging setting of generalizing to unseen graphs as well.

**Model architecture.** We identify each vertex and edge in the road network with a unique token ID and learn an embedding table to map these IDs to embedding vectors. Although this approach is not scalable, as it requires the training dataset to contain enough information about every vertex and edge to populate an embedding table, we maintain this naive approach for illustrative purposes and present an efficient solution in Section 3.

Our proposed model architecture is illustrated in Figure 2 and consists of four components. The first component tokenizes the queries and passes the token IDs through a learned embedding table. The second component assigns token IDs to the source vertex, destination vertex, and the vertices in the incomplete route (prefix), and passes the token IDs through a second learned embedding table. The sequences of embedding vectors output by these two components are concatenated into a long sequence of vectors and passed into a causal decoder only transformer, which we will refer to as the *base network* and which consists of stacked Transformer blocks that each contain a multi-layer perceptron (MLP), multi-head attention layer, and residual connection (Brown et al., 2020).

The output of the base network is also a sequence of vectors: we refer to the vector at the first position of the sequence as the *problem embedding*, interpreting the vector as encoding all necessary information about the customized routing task and the incomplete route so far. To make a next-vertex prediction, a fourth component takes the token IDs of candidate vertices and passes the token IDs into an embedding table. The candidate vertices are simply the vertices in the graph that are connected to the last vertex in the prefix so far. The model's prediction distribution over these

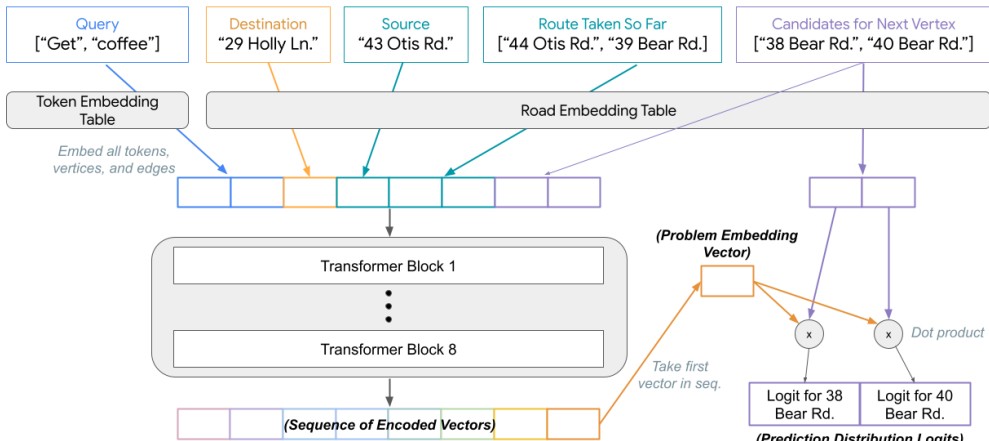

Figure 2: Architecture of the overall model for next vertex prediction.

candidates is then obtained by defining the logit for each candidate as the dot product between the candidate's embedding vector and the problem embedding vector. We then train the model's parameters, including all of the embedding tables and base network parameters, by minimizing a cross-entropy loss.

**Inference algorithms.** Given a next-vertex prediction model and a customized routing problem, there are several approaches to obtaining a candidate route. The first approach is greedy decoding, which is a popular baseline approach in language modeling as well (Chowdhery et al., 2022). This is computationally quite efficient for the routing setting. To observe this note that an important advantage of our next-vertex prediction framework is that run time complexity is independent of the size of the road network. Rather, a single next-vertex inference step requires only running time that is quadratic in the length of the route so far, implying a run time complexity that is cubic in route-length $O(|\text{Route}|^3)$ for greedy decoding, with improved complexity possible through more efficient attention mechanisms (Wang et al., 2020; Zaheer et al., 2020; Dao et al., 2022).

We also take inspiration from natural language generation and apply beam search (Cohen & Beck, 2019) to our next-vertex prediction model. Indeed, beam search appears to be empirically the most effective inference algorithm and, by tuning the beam width, allows for one to cleanly trade-off accuracy for inference complexity, with a run time of $O(\text{BeamWidth}^2 \cdot |\text{Route}|^3)$. We can also define analogues of classical graph algorithms that treat the logits of a next-vertex prediction model as pseudo-edge-weights. For example, in Appendix B.10, we study a next-vertex analogue of Dijkstra's algorithm, finding that it can offer certain performance gains when the underlying next-vertex prediction model is of poor quality.

**Experiment setup.** In our experiments, depicted in Table 1, we train next-vertex prediction models on 20 million next-vertex prediction data points, reflecting approximately 1 million examples of customized routing tasks and their ground-truth routes. These customized routing tasks feature queries that contain 3-4 logical clauses; an example of a query with 3 logical clauses is asking for a route that passes by a coffee shop (clause 1) and either a lake (clause 2) or a dog park (clause 3).

We study two different datasets of customized routing tasks. In the first dataset, which is indicated by the "Template-Generated Queries" column, we use a simple sentence-template to programatically generate a corpus of 79,927 different query texts, which we use in conjunction with a simulated road network to construct examples of customized routing tasks. In the second dataset, which is indicated by the "LLM-Generated Queries" column, we instead construct a diverse corpus of 913,453,361 natural language queries using the commercial large-language-model GPT-4 (OpenAI, 2023). We defer the details to Appendix C.

**Methods and baselines.** The first three rows of Table 1 depict the routing performance when using beam-search (width 10) to unroll next-vertex prediction models. These performance metrics

| | Template-Generated Queries | LLM-Generated Queries |
|---|---|---|
| Fulfills Query (Model) | $91.8\% \pm 0.7\%$ | $88.8\% \pm 0.7\%$ |
| Excess Travel Time (Model) | $3.5\% \pm 0.3\%$ | $3.2\% \pm 0.1\%$ |
| GT Route Recovery (Model) | $64.7\% \pm 0.9\%$ | $63.8\% \pm 0.6\%$ |
| Fulfills Query (EF Baseline) | $95.1\%$ | $95.0\%$ |
| Excess Travel Time (EF Baseline) | $39.9\%$ | $39.7\%$ |
| GT Route Recovery (EF Baseline) | $7.0\%$ | $6.1\%$ |

Table 1: Performance of next-vertex prediction models on a dataset of previously unseen customized routing tasks on a 625-vertex grid network. The left column reports metrics on customized routing tasks constructed from a corpus of queries generated by a simple sentence template. The right column reports metrics on tasks constructed from large-language-model (LLM) generated queries. (Model) denotes the performance of width-10 beam search on a next-vertex prediction model. (EF Baseline) denotes the performance of sampling 512 candidate routes using electrical flows and omnipotently chose the best one. Results are aggregated across 5 random seeds, with standard error denoted by $\pm$. GT Route Recovery is the probability of exactly recovering the ground-truth route.

are measured on customized routing tasks that are excluded from the training data; in particular, for every customized routing task $(s, u, v)$ in the test set, the train set is guaranteed to neither have any tasks that share the same source-destination pair $u, v$, e.g. $(s', u, v)$, nor any tasks that share the same (or even semantically similar) query $s$.

For comparison, the bottom three rows in Table 1 depict the performance of a baseline algorithm that uses unguided search via electrical flows (EF) as described in (Sinop et al., 2021) to generate 512 unique candidate routes and an omnipotent referee to pick the best route out of the 512 candidates.

As existing routing algorithms in literature are inapplicable to the very general customized routing tasks explored in this paper, combining electrical flows with an omnipotent referee provides a strong baseline that underscores how non-trivial these routing tasks are. We emphasize that a practical implementation of the EF baseline will perform significantly worse than the performance reported in Table 1, as generating and evaluating 500+ candidate routes on massive graphs is impractical and expensive —more importantly—the baseline assumes access to an omnipotent referee.

**Results.** In Table 1, we observe that the trained next-vertex prediction model fulfills $\sim 90\%$ of previously unencountered queries with only a marginal $3\%$ increase in travel time. Furthermore, somewhat surprisingly, the model *exactly* recovers the ground-truth route over $60\%$ of the time. We also note that—across the board—the next-vertex prediction model's performance metrics are better by a statistically significant margin on the dataset with less challenging queries ("Template-Generated Queries"). This underscores how natural language processing contributes to the difficulty of customized routing, which involves both NLP and routing.

In comparison, the EF Baseline performs significantly worse, recovering the ground-truth route $6\%$ of the time. In other words, on $94\%$ of routing tasks, among the 512 unique routes obtained by the EF algorithm from the source to the destination, none were the ground-truth. In contrast, $63\%$ of the time, the single route outputted by the next-vertex prediction model is the ground-truth.

We note that, because we required the EF baseline's omnipotent referee to prioritize fulfilling queries over minimizing travel time, the baseline achieves a higher rate of query fulfillment than the next-vertex prediction model, but at the cost of increasing travel times by $40\%$. When we similarly require our next-vertex prediction model to prioritize fulfilling queries, the model can achieve a similar fulfillment rate of 95% (Figure 22).

## 3 SCALING TO BIG ROAD NETWORKS

In this section, we scale our proposed next-vertex prediction framework to large road networks, including the road network of the United States (OpenStreetMap contributors, 2017).

### 3.1 FROM EMBEDDING TABLES TO ROAD EMBEDDING NETWORKS

One shortcoming of the model architecture described in Section 2 is that learning embeddings for every vertex and edge in the road network is impractical at real-world scale. To address this, we move away from identifying vertices and edges with unique token IDs. Instead, we will identify vertices and edges with features that encode information about the topology of their local neighborhood of the road network—which we will refer to as their receptive field—and the points-of-interest that they are co-located with. These features are not necessarily vectors; in our proposed featurization scheme, depicted in Figure 3, vertex and edge features are sequences of vectors, where each entry in the sequence corresponds to a vertex in the receptive field.

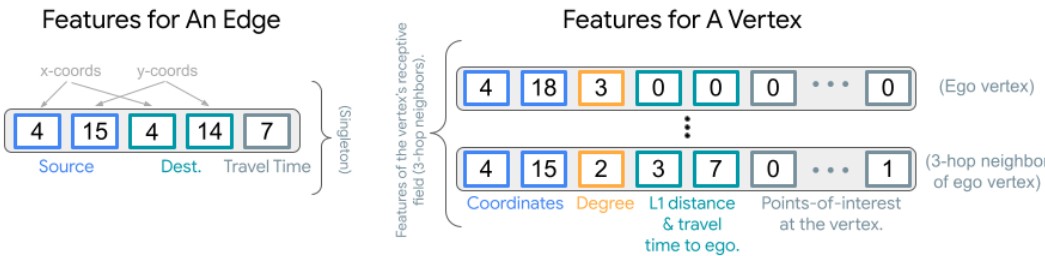

Figure 3: Features for vertices and edges, which are given as input to road embedding networks.

**Road embedding networks.** Having replaced vertex and edge token IDs for vertex/edge features, we now replace the vertex and edge embedding tables described in the previous section with embedding networks. Because we produce vector sequences as the features for a vertex/edge, we introduce a second smaller BERT-style model (Devlin et al., 2018) as our road-embedding network. This network first applies several blocks of MLPs, independently to each vector in the feature vector sequences, then applies transformer blocks to the resulting sequence of vectors, and finally collapses the vector sequence into a single embedding vector at the output of the road embedding network.

### 3.2 SECONDARY SCORING MODEL.

We next introduce another important piece, the *scoring module*, that improves the performance of our models during inference. As we discussed in Section 2, many inference algorithms for "unrolling" next-token prediction models can produce a set of candidate sequences. For example, one can produce $m$ candidate sequences by running beam search with width $m$ and returning all sequences within the output beam instead of choosing the best sequence. The typical practice in domains such as language modeling is to then output the sequence in the beam to which the next token prediction model assigns the highest probability (Brown et al., 2020; Chowdhery et al., 2022).

We instead find that for routing problems, especially at large scales, having beam search return a set of candidate routes and training a secondary model to choose the best route among them can significantly increase performance. Hence, for the experiments in this section, we implement a secondary scoring model that, given a customized routing task and a candidate route, uses a fully-trained next-vertex prediction model to obtain a "problem embedding", and passes the problem embedding through a one-hidden-layer neural network to obtain a scalar score.

In order to train the scoring model, we use a fraction of our training set to create pairs of positive and negative routes. For a (*source*, *destination*) pair, the ground truth route serves as the positive example and a random path from the source to the destination (sampled from the distribution of the next-vertex prediction model) serves as the negative pair. We embed these routes via the trained next-vertex prediction model and train an MLP applied to the embeddings to assign higher scores to the positive example via a simple logistic loss function. After training the scoring model, at inference time, we select from candidate routes by choosing the route with the highest such score.

|                                   | Prev. Unseen Roads | Prev. Unseen Queries & Unseen Roads |
|-----------------------------------|--------------------|-------------------------------------|
| Fulfills Query (Model)            | 94.7% ± 0.5%       | 90.2% ± 0.7%                        |
| Excess Travel Time (Model)        | 2.8% ± 0.2%        | 3.5% ± 0.2%                         |
| GT Route Recovery (Model)         | 68.4% ± 0.8%       | 54.9% ± 1.2%                        |
| Fulfills Query (EF Baseline)      | 77.5%              | 61.3%                               |
| Excess Travel Time (EF Baseline)  | 10.9%              | 8.4%                                |
| GT Route Recovery (EF Baseline)   | 23.0%              | 7.3%                                |

Table 2: Performance on the road networks of New Hampshire, Rhode Island, and Mississippi (all excluded from training data). The tasks feature either previously seen (left) or previously unseen queries (right). (Model) denotes the performance with width-10 beam search and a secondary scoring model. (EF Baseline) denotes the performance of the best of 512 candidate routes sampled with electrical flows. Results are aggregated across 5 random seeds, with standard error denoted by ±. GT Route Recovery is the probability of exactly recovering the ground-truth route.

## 3.3 Experiments

We now consider customized routing tasks on the, very large, United States road network (Table 2) and a simulation of an infinitely large grid road network (Table 3).

**Experimental setup.** We train next-vertex prediction models on 100 million datapoints, reflecting approximately 5-10 million examples of customized routing tasks and their ground-truth routes. These routing tasks build on the same corpus of 913,453,361 large-language-model-generated queries described in Section 2.1. We sample the source-destination pairs for these tasks by randomly choosing 256-vertex subgraphs of each road network, choosing a source and destination from this subgraph, and finding a ground-truth route that routes exclusively through this subgraph. In Tables 2 and 3, (Model) rows report the performance of applying beam-search of width 10 to fully trained next-vertex prediction models and choosing a final route using the secondary scoring model. The (EF Baseline) rows report the performance of a baseline algorithm that uses unguided search via electrical flows to generate 512 unique candidate routes and an omnipotent referee to pick the best.

Our experiments (Table 2) on the United States road network build on map data from the OpenStreetMap repository (OpenStreetMap contributors, 2017), with light pre-processing so that routing tasks are non-trivial, e.g., contracting vertices of degree two since one can only go straight on such roads. The reported metrics are evaluated on customized routing tasks on road networks in *Rhode Island, New Hampshire, and Mississippi*, whose road networks are wholly omitted from the training data. Our experiments (Table 3) on the infinite grid road network build on the same construction as the experiments in Section 2.1. The reported metrics are evaluated on subgraphs of the grid road network that are wholly omitted from training data.

In both Tables 2 and 3, the left column reports metrics for tasks involving queries which have been previously observed (albeit on different roads) and the right column reports metrics for tasks involving queries that have never been previously observed (even including semantically similar queries). We note that the query train set and query test set are not identically distributed, as we constructed the query test set to be more difficult. For example, every customized routing task in the right column features at least two logical clauses in each query.

**Results.** In Tables 2 and 3, we observe that next-vertex prediction models achieve over 95% fulfillment rate on previously unseen queries on large grid road networks and over 90% fulfillment on the United States road network. Excess travel time is marginal, at less than 2% and 4% respectively. Interestingly, these models still exactly recover the ground-truth routes over 50% of the time on the US road network. In contrast, the electrical flows baseline significantly underperforms in all dimensions. In other words, even the best of the 512 routes sampled by electrical flows leads to an explosion in travel time and a significant query non-fulfillment rate: 24% vs next-vertex prediction's 5% on grid road networks, and 45% vs next-vertex prediction's 10% on US road networks.

**Varying beam width, varying electrical flows.** We can examine the trade-off between the performance and efficiency of our next-vertex prediction model by varying the width of the beam-search.

|                                 | Prev. Unseen Roads | Prev. Unseen Queries & Unseen Roads |
|---------------------------------|--------------------|-------------------------------------|
| Fulfills Query (Model)          | $98.3\% \pm 0.1\%$ | $95.1\% \pm 0.2\%$                  |
| Excess Travel Time (Model)      | $1.3\% \pm 0.1\%$  | $1.7\% \pm 0.0\%$                   |
| GT Route Recovery (Model)       | $86.3\% \pm 0.3\%$ | $76.4\% \pm 0.5\%$                  |
| Fulfills Query (EF Baseline)    | $87.0\%$           | $84.2\%$                            |
| Excess Travel Time (EF Baseline)| $26.3\%$           | $28.9\%$                            |
| GT Route Recovery (EF Baseline) | $18.8\%$           | $8.7\%$                             |

Table 3: Performance on 256-vertex subgraphs of a simulated grid road network (wholly excluded from training data). The tasks feature either previously seen (left) or previously unseen queries (right). (Model) denotes the performance of width-10 beam search and the secondary scoring model. (EF Baseline) denotes the performance of the best of $512$ candidate routes sampled with electrical flows. Results are aggregated across 5 random seeds, with standard error denoted by $\pm$. GT Route Recovery is the probability of exactly recovering the ground-truth route.

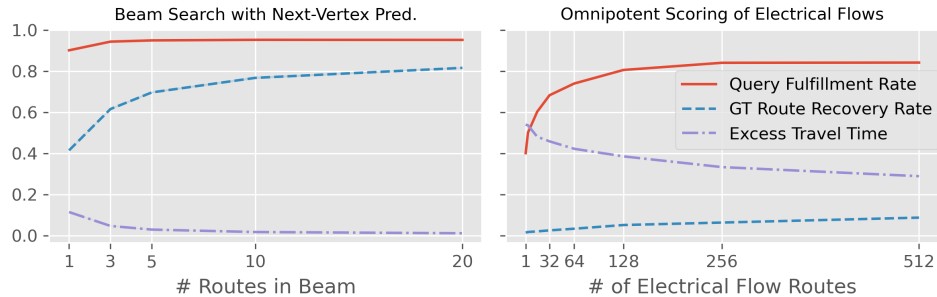

Figure 4: Performance on 256-vertex subgraphs of a simulated grid road network on previously unseen roads and queries. The left plot shows the performance of applying beam search with various widths. The right plot shows the performance of the best of $k$ candidate routes sampled with electrical flows for various choices of $k$. Results are averaged over 5 seeds, with standard error denoted by lighter colors.

In Figure 4, we evaluate the same next-vertex prediction models from Table 3, again on previously unencountered queries and roads from an infinite grid road network, but with varying beam widths. We observe monotonic improvements in all metrics with increasing beam-width, including improving upon the metrics reported in Table 3 by increasing beam width to 20. As a comparison, we plot on the left the performance of the best of the $k$ routes sampled by the electrical flow algorithm, observing significantly poorer scaling behavior when varying $k$ from 1 to $512$ than compared to scaling beam search width. We depict an analogous experiment in Figure 21 on the next-vertex prediction models trained on the US road network.

**Additional results.** We include in the Appendix additional experiments that study the relationship between model performance and (1) how much of the road network the models observe (Figure 11), (2) the scale of the models' network architectures (Figure 12, 13), (3) the difficulty of a dataset's route customization queries (Figure 14), and (4) one's choice of inference algorithm (Figure 23). We also perform ablation studies of the road embedding network (Figure 17) and secondary scoring model (Figure 22). We also visualize and analyze the attention matrices of our networks (Figures 15, 16) and the road embeddings that our networks learn (Figure 18).

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

## A  EXAMPLE OF PREDICTED ROUTES

This section provides examples of customized routing tasks on previously unseen subgraphs of the United States road network (OpenStreetMap contributors, 2017) and simulated grid road networks. These examples are sampled randomly, without cherry-picking, and depict a customized routing task, the ground-truth route, and our model's predicted route. The models used to generate the route predictions are the same models from the experiments depicted in Tables 2 and 3. We emphasize that, during training, these models have never encountered the subgraphs of the road networks that are represented in these tasks, nor have the models encountered queries with these syntax or semantics before.

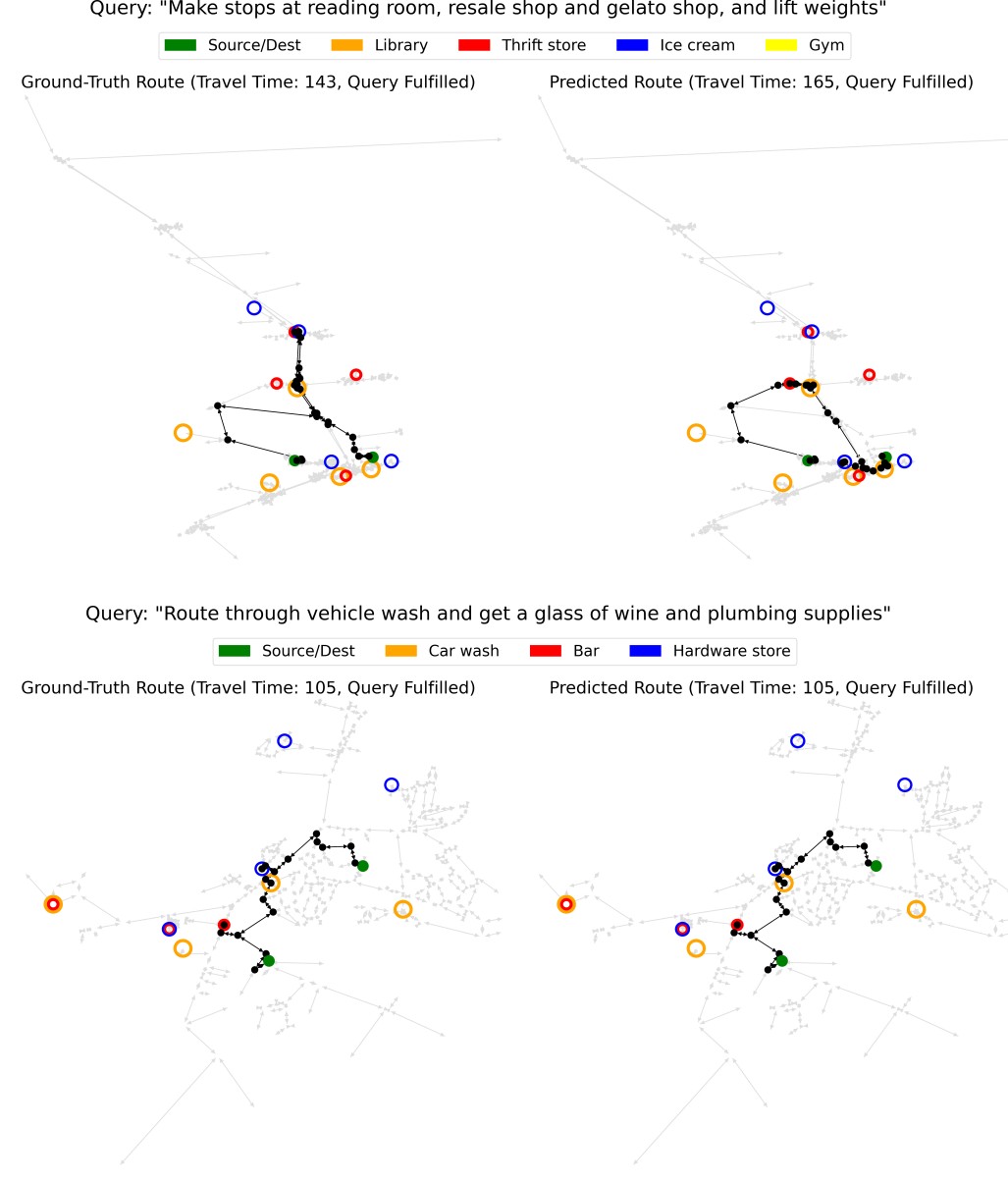

Figure 5: Examples of customized routing tasks on road networks from New Hampshire, Mississippi and Rhode Island. Grey elements denote the road network and green dots denote source and destination. Black elements denote the ground-truth route on the left and our model's predicted route on the right. Colored circles indicate points-of-interest along the road network.

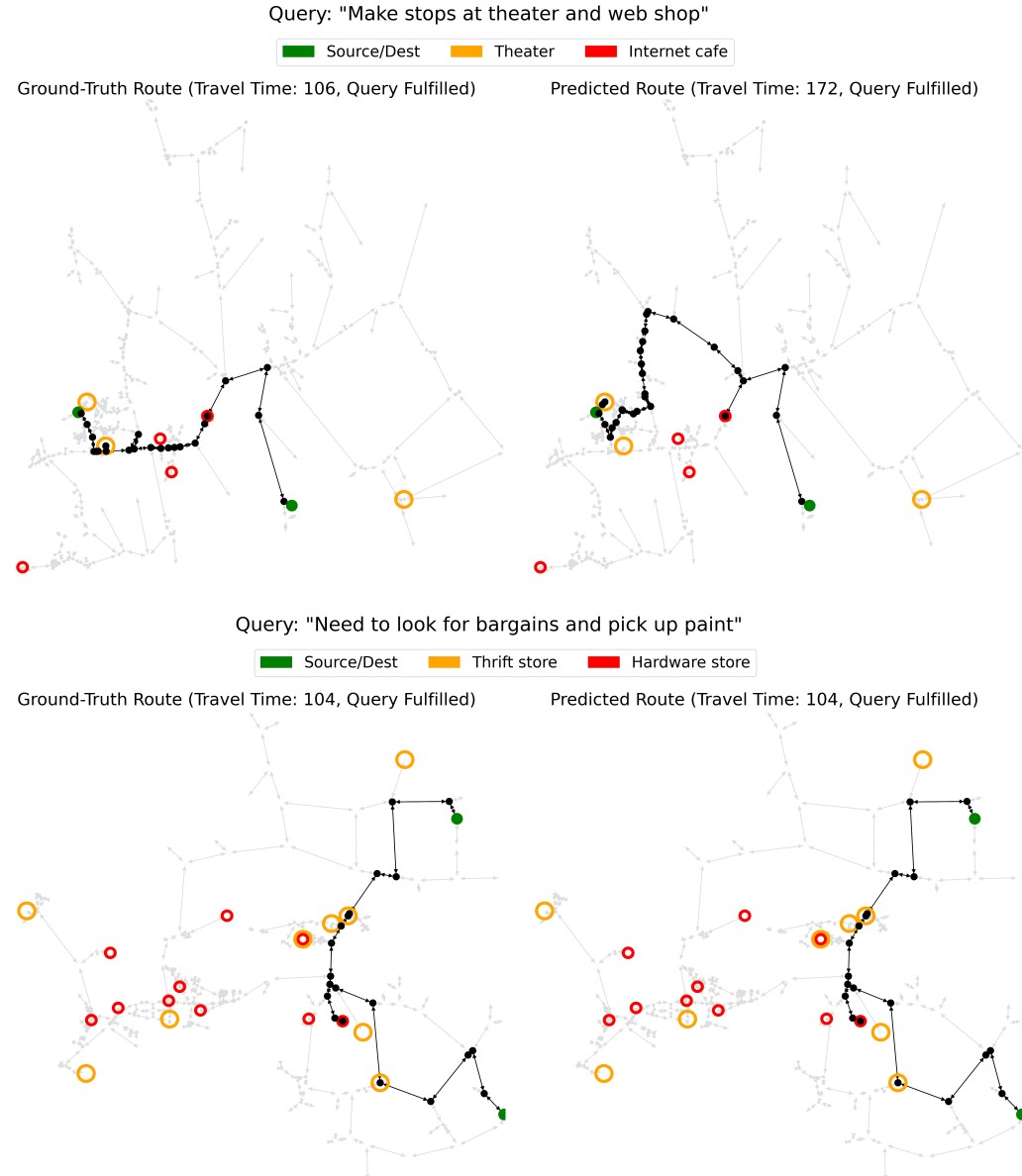

Figure 6: Further examples of customized routing tasks on road networks from New Hampshire, Mississippi and Rhode Island. See Figure 5.

Figures 5, 6 and 7 depict customized routing tasks on real-world road networks within the United States, taken from OpenStreetMap data (OpenStreetMap contributors, 2017). These road networks belong to the states of New Hampshire, Missisipi, and Rhode Island—states that the models are not trained on.

Figures 8, 9 and 10 depict customized routing tasks on previously unseen subgraphs of a large grid road network.

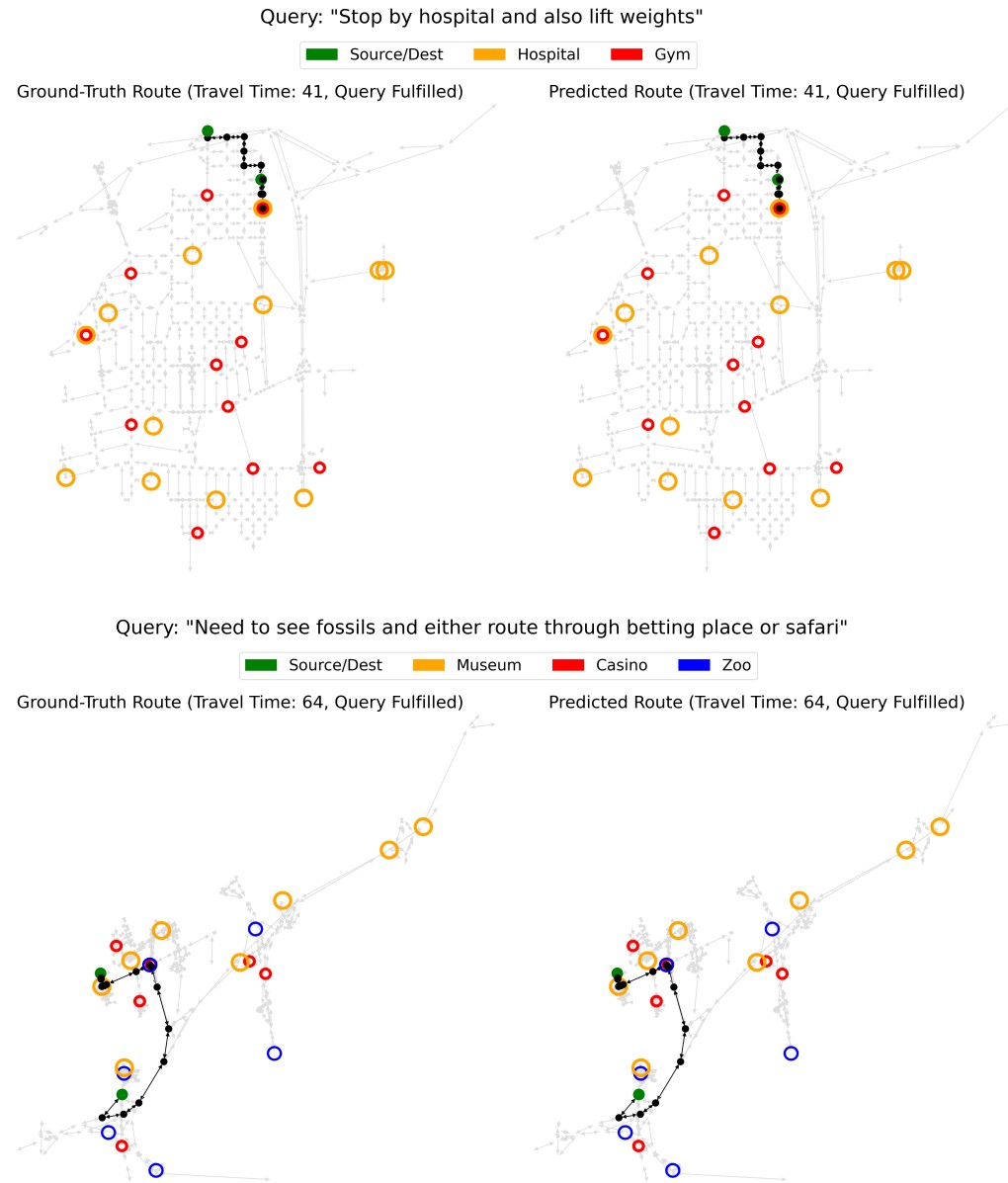

Figure 7: Further examples of customized routing tasks on road networks from New Hampshire, Mississippi and Rhode Island. See Figure 5.

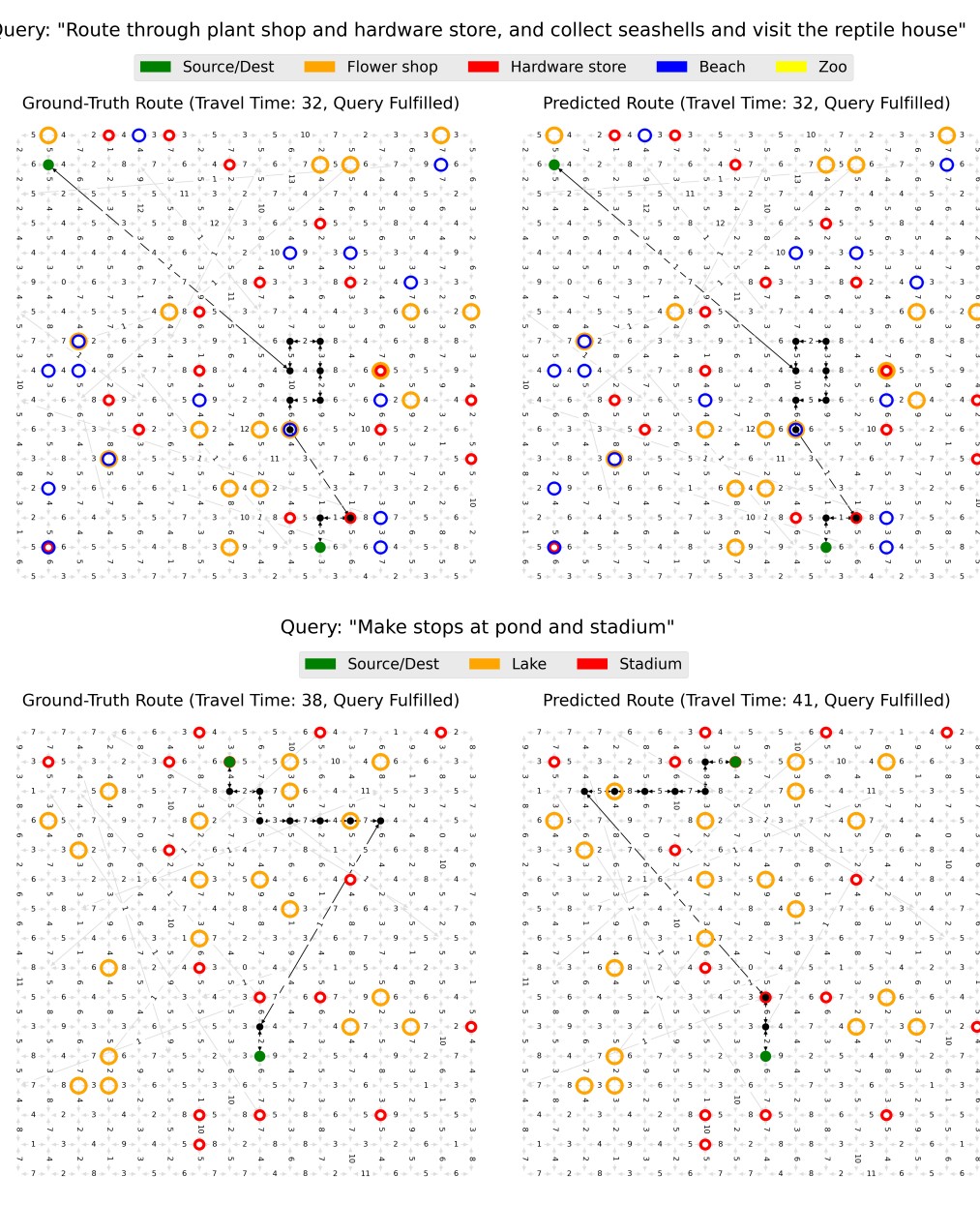

Figure 8: Examples of customized routing tasks on previously unseen subgraphs of a large grid road network. Grey elements denote the road network, with number annotations indicating travel times along each road. Green dots denote source and destination. Black elements denote the ground-truth route on the left and our model's predicted route on the right. Colored circles indicate points-of-interest along the road network.

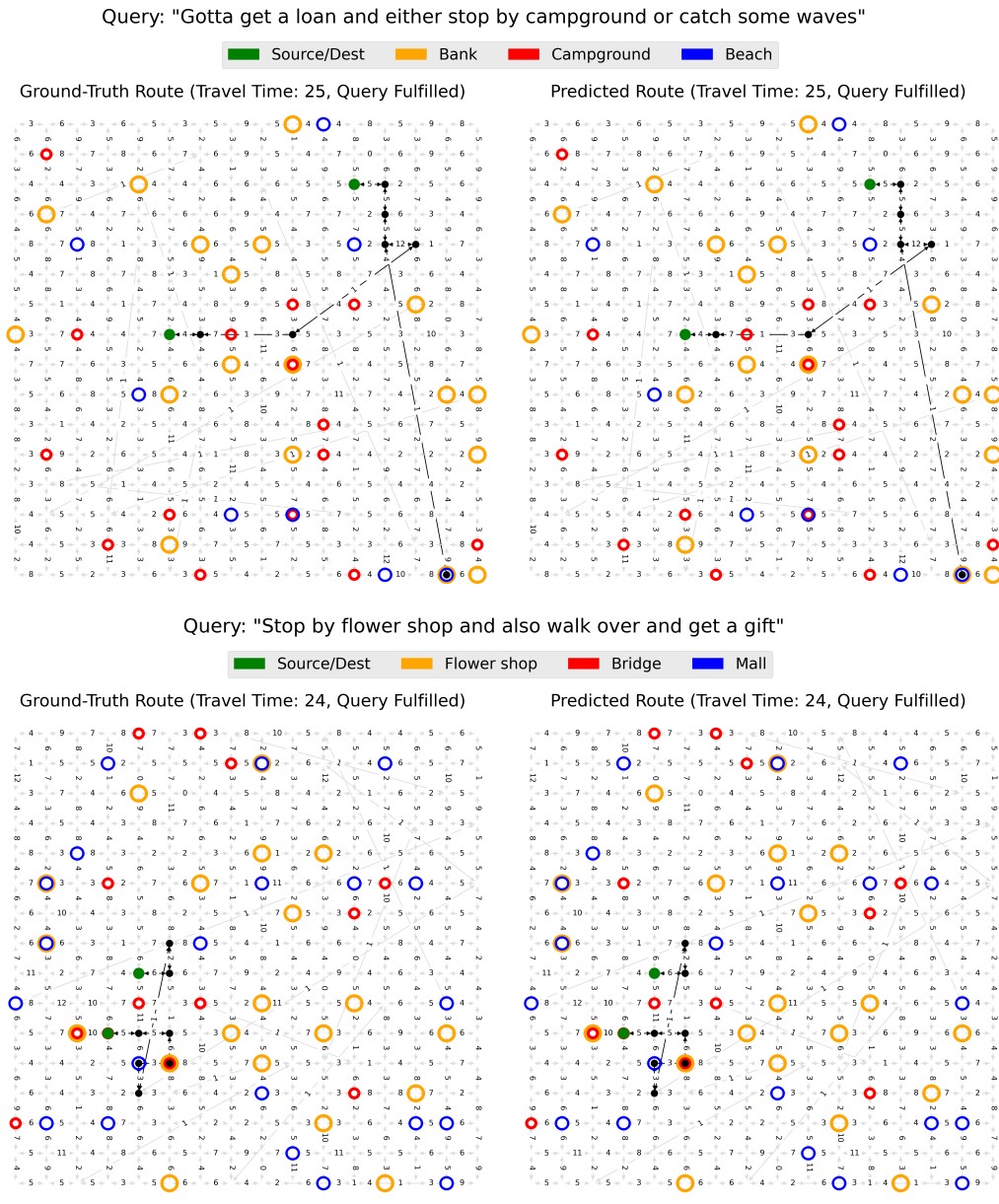

Figure 9: Further examples of customized routing tasks on previously unencountered subgraphs of a large grid road network. See Figure 8.

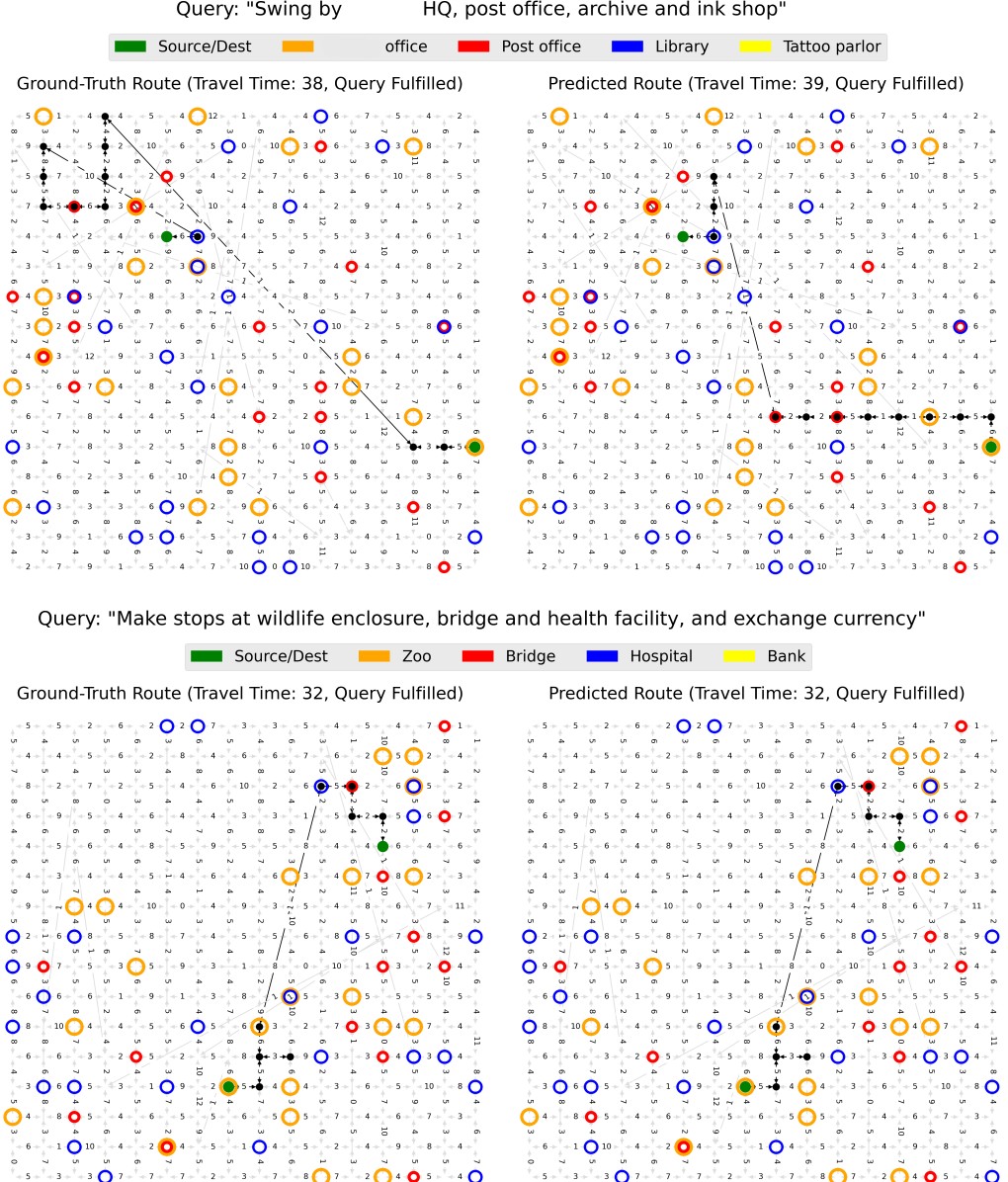

Figure 10: Further examples of customized routing tasks on previously unencountered subgraphs of a large grid road network. See Figure 8.

Next-Vertex Accuracy vs Receptive Field of Road Features

Figure 11: Plot of next-vertex prediction accuracy versus $k$, where $k \in \mathbb{Z}_+$ determines the $k$-hop neighborhood that constitutes the receptive field of road features. Smaller values of $k$ mean that the model can only observe a smaller subgraph of the road network during inference-time.

## B ADDITIONAL EXPERIMENTS

In this section, we detail additional experiments to better understand the challenges of customized routing and the behaviors of our next-vertex prediction models. We note that the experiments in this section are conducted on a smaller scale than the experiments depicted in Tables 2 and 3, as the goals of the experiments in this section are deeper insights rather than state-of-art performances. In particular, these experiments are conducted with smaller network architectures, smaller datasets consisting of 20,000,000 examples, and fewer training iterations. As a result, the performance metrics reported in this section are not directly comparable to those of Tables 2 and 3.

### B.1 RECEPTIVE FIELD OF ROAD FEATURES

In this experiment, plotted in Figure 11, we revisit how much of the road network we allow our next-vertex prediction models to observe during inference-time and the relationship between this choice and the performance of our models. Recall that, during inference time, our next-vertex prediction models can only observe the road network through the input sequence we provide it with (e.g., the destination, source, and roads taken so far). In addition, the road features we provide for each vertex and edge in our input sequence also includes information about their $k$-hop neighborhood, which we refer to as their *receptive field*. Whereas we usually fix $k = 3$ in the rest of our experiments, in this experiment we vary $k$ from a value of 1 to a value of 4.

**Experiment details.** This experiment is conducted on large grid road networks and our dataset of LLM-generated points-of-interest queries. While these experiments vary the value of $k$—that is, the $k$-hop neighborhood constituting the receptive field of our road features—all other parameters are kept constant and stated in full in Table 6. We note that the value of $k$ that we fix also sets a hard cutoff on the size of each receptive field: when the $k$-hop neighborhood of a vertex exceeds $4 \cdot k^2$ vertices, we only allow the vertex's features to encode the $4 \cdot k^2$ closest vertices in the neighborhood.

**Observations.** Limiting our next-vertex prediction models to $k = 1$—that is, observing only a very local subgraph—significantly degrades their performances. There is also a statistically significant improvement yielded by increasing $k = 2$ to $k = 3$. However, increasing the size of the receptive fields from $k = 3$ to $k = 4$ no longer improves training error convergence by a statistically significant amount—rather it negatively impacts test accuracy. The latter observation is likely because setting $k = 4$ involves training a model on significantly more complicated inputs, contributing

Figure 12: Plot of next-vertex prediction accuracy for networks of varying embedding dimensions. A larger embedding dimension means that a network has more free parameters and representation power.

to training instability. These observations highlight the role of $k$ as an important hyperparameter that allows one to reliably trade-off inference efficiency with accuracy.

### B.2 SCALING STUDIES

In this experiment, we implement next-vertex prediction models for network architectures of varying sizes, following conventional rules-of-thumb for scaling large language models. In particular, we study the performance of our next-vertex prediction models as we scale our network architectures in two directions: the embedding dimensions of the transformer blocks (Figure 12), and the size and depth of the road embedding networks (Figure 13).

**Experiment details.** This experiment is conducted on large grid road networks and our dataset of LLM-generated points-of-interest queries. The experiment consists of two parts.

In the first part, we vary the width of the transformer blocks in our network architecture, increasing the embedding dimension of the base network from $64 \rightarrow 256 \rightarrow 512 \rightarrow 1024$. We follow the canonical scaling strategy for large language models and correspondingly increase the depth of the base network from $4 \rightarrow 6 \rightarrow 8 \rightarrow 8$ transformer blocks. Similarly, we scale the intermediate dimension of the base network from $256 \rightarrow 512 \rightarrow 1024 \rightarrow 2048$ and the number of attention heads in the base network from $4 \rightarrow 4 \rightarrow 8 \rightarrow 8$. We also scale the embedding and intermediate dimensions of the transformer blocks in the road embedding network from $64 \rightarrow 64 \rightarrow 128 \rightarrow 128$ and from $128 \rightarrow 256 \rightarrow 256 \rightarrow 512$ correspondingly. These parameter values, and others, are specified in detail in Table 9. The results of these networks are depicted in Figure 12.

In the second part of this experiment, we scale up the depth of the road embedding network, increasing the number of transformer blocks in the road embedding network from $1 \rightarrow 2 \rightarrow 4 \rightarrow 4$ and the number of additional MLP blocks from $1 \rightarrow 2 \rightarrow 2 \rightarrow 4$. We correspondingly increase the number of attention heads in the road embedding network from $1 \rightarrow 2 \rightarrow 4 \rightarrow 4$. All other parameters, including embedding dimension, are kept constant; they are specified in full in Table 8. The results of these networks are depicted in Figure 13.

**Observations.** Along both of the axes in which we attempt to scale up our models, we observe that scaling up leads to both improved train error and test error convergence until hitting a saturation point. Past the saturation point, train error and test error convergence no longer increase with scale. However, we also do not observe a statistically significant decrease in test performance due

Next-Vertex Prediction Accuracy vs Road Embedding Network Depth

Figure 13: Plot of next-vertex prediction accuracy for networks with road embedding subnetworks of varying depth. "X MLP + Y Transformers" indicates the road embedding subnetwork consists of X multi-layer perceptron blocks followed by Y transformer blocks.

to generalization error, even past the saturation point. This is in line with previous observations of overparameterization in language models.

### B.3 POINTS-OF-INTEREST DENSITY

In this experiment, plotted in Figure 14, we attempt to understand how the difficulty of fulfilling a query affects the difficulty of next-vertex prediction. In particular, consider the difficulty of fulfilling the queries in our dataset, which asks for routes with short travel time but that stop by certain points-of-interest along the way. One should expect that, when points-of-interest are extremely plentiful, customized routing reduces to finding the shortest path since the shortest path likely encounters the desired points-of-interest by chance. That is, asking for the shortest route that stops by a coffee shop is practically the same as asking for the shortest route. In this experiment, we train next-vertex prediction models on multiple variants of our customized routing tasks, where we vary the probability that a vertex in the road network contains a point-of-interest thus varying the density of points-of-interest on our road networks.

**Experiment details.** This experiment is conducted on large grid road networks and our dataset of LLM-generated points-of-interest queries. In this experiment, we increase the default probability of a vertex being a point-of-interest from 0.05% to either 0.5% or 5%. This probability is usually set to 0.05% in our grid road networks and 0.005% on the United States road network. All other parameters are kept constant and stated in full in Table 7.

**Observations.** A high point-of-interest density of 5% indeed corresponds to a significantly easier learning task, as indicated by the significantly improved error curves. Interestingly, decreasing the density of points-of-interest from 0.5% to 0.05% does not significantly affect the difficulty of the next-vertex prediction problem. We attribute this to the fact that, once points-of-interest density is low enough that one needs to go out of their way to find points-of-interest, any further decrease in density only affects the difficulty of routing to the extent that one needs to travel further to find desired points-of-interest. Moreover, decreasing density also makes the routing problem easier in the sense that, given a point-of-interest category (e.g., coffee shop), there are fewer points-of-interest in the category to choose from (fewer coffee shops in the area to choose between).

Next-Vertex Accuracy vs Points-of-Interest Density

Figure 14: Plot of next-vertex prediction accuracy versus the density of points-of-interest on the road network. Greater density indicates more plentiful points-of-interest on the road network, affecting the difficulty of routing tasks where queries impose constraints on visitation to points-of-interest.

### B.4 VISUALIZING ATTENTION MATRICES

In this section, we visualize the attention weights of the transformer blocks in fully trained next-vertex prediction models. These visualizations, depicted in Figures 15 and 16, show the attention weights of transformers in the base network and the road embedding network, respectively, on a randomly sampled datapoint.

**Attention weights in the base network.** Figure 15 depicts the attention weights of the third transformer block in the base network of a fully trained next-vertex prediction model that is given a randomly sampled next-vertex prediction problem. Here, we have averaged the attention weights across every attention head. The axes in Figure 15 are annotated with the semantic values of each position in the input sequence. The "Destination" position contains the embedding for the destination vertex. The positions annotated with "want", "to", "need", "some", "med", "# #s", "and", "go", "to", "confession" contain the embeddings for the tokens in the query "Want to need some meds and go to confession". The "[SEP]" positions contain a placeholder separation token. The "Candidate" positions contain embeddings for the vertices that are candidates to be chosen as the next vertex in the route. The "Source" position contains the embedding of the source vertex, while the subsequent positions contain the embeddings of the edges and vertices that have already been taken in the route. Two of the positions are annotated with "[Pharmacy]" and "[Church]", indicating those vertices already contained the points-of-interest "pharmacy" and "church".

We can visually confirm several interesting behaviors in the attention matrix. First, we can see the network places the most weights on the query tokens "med", "and", and "confession". These tokens contain the query's two point-of-interest constraints and their logical conjunction, confirming the network is able to extract key information from text queries. On a similar note, we can see an abnormally strong weight with the "confession" token as the query and the vertex that contains a church point-of-interest as the key. This indicates the network both recognizes that "confession" is indicating a church point-of-interest and recognizes that the route has already visited a church. The checkerboard pattern in the bottom right corner can be attributed to the network attending vertex positions to vertex positions, and edge positions to edge positions.

**Attention weights in the road embedding network.** Figure 16 depicts the attention weights of the first transformer block in the road embedding network of a fully trained next-vertex prediction model that is given a randomly sampled vertex from a grid road network. Here, we have averaged the attention weights across every attention head. The axes in Figure 16 are annotated with the semantic

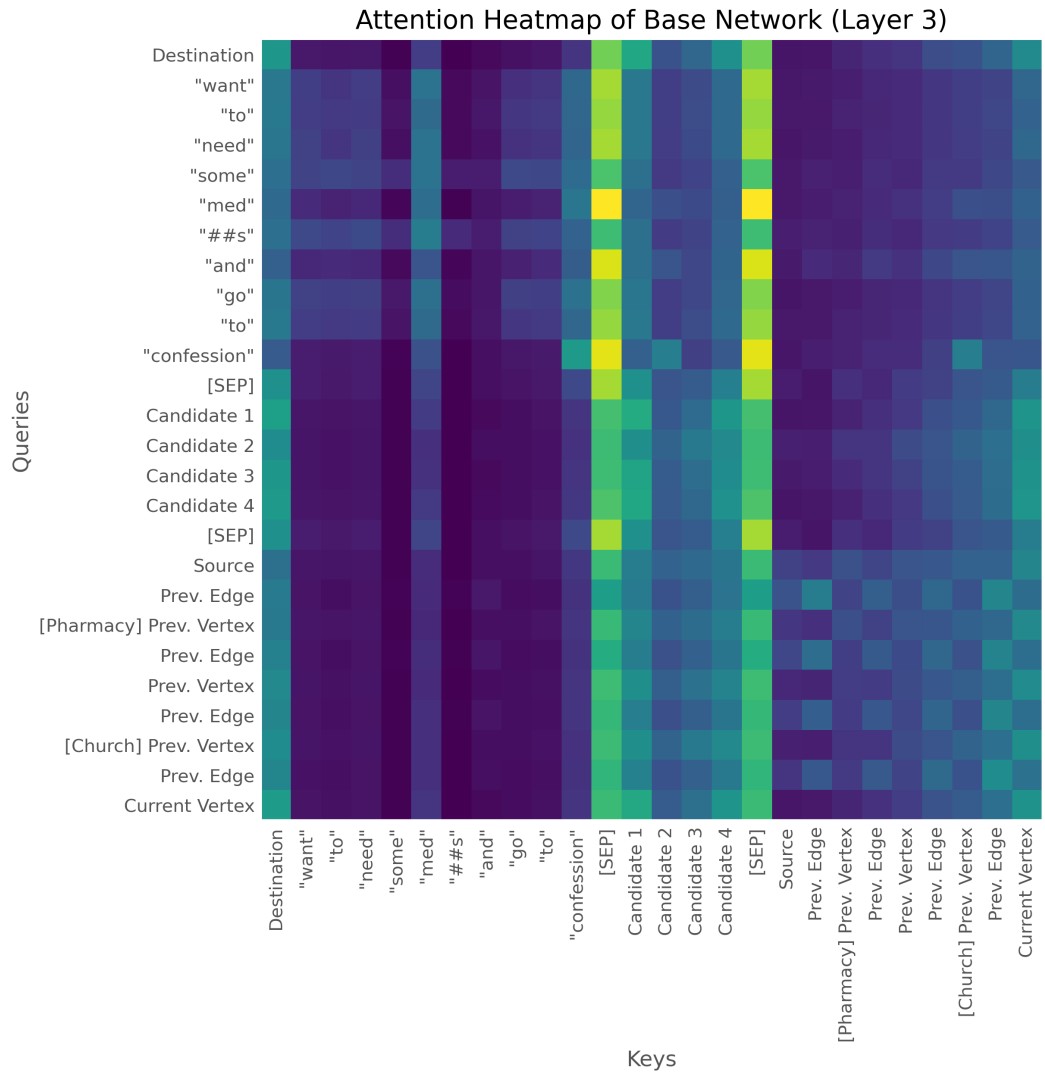

Figure 15: The attention weight matrix of the third transformer block in base embedding network of a fully trained next-vertex prediction model. The model was given a randomly sampled next-vertex prediction problem as input. "Destination", "Source", and "Candidate x" positions contain embeddings for the destination, source, and candidate vertices respectively. The positions "want", "to", "need", "some", "med", "# #s", "and", "go", "to", "confession" contain embeddings for the query. The remaining positions represent embeddings of vertices and edges already visited in the route, with the annotations "[Church]" and "[Pharmacy]" indicating vertices that contain the points-of-interest church and pharmacy respectively. Warmer colors indicate greater weight.

value of each position in the sequence. "Ego" denotes that a position in the feature sequence that encodes information about the vertex that the road embedding network was given as input. "Neighbor" denotes a position in the sequence that encodes information about an immediate neighbor to the ego vertex, while $k$-hop denotes a position encoding information about a $k$-hop neighbor to the ego vertex.

We can observe that, although the transformer treats each position in the sequence symmetrically, the transformer places the most weight on the ego vertex as a key than as a query and significantly more weight on immediate neighbors than 2-hop or 3-hop neighbors.

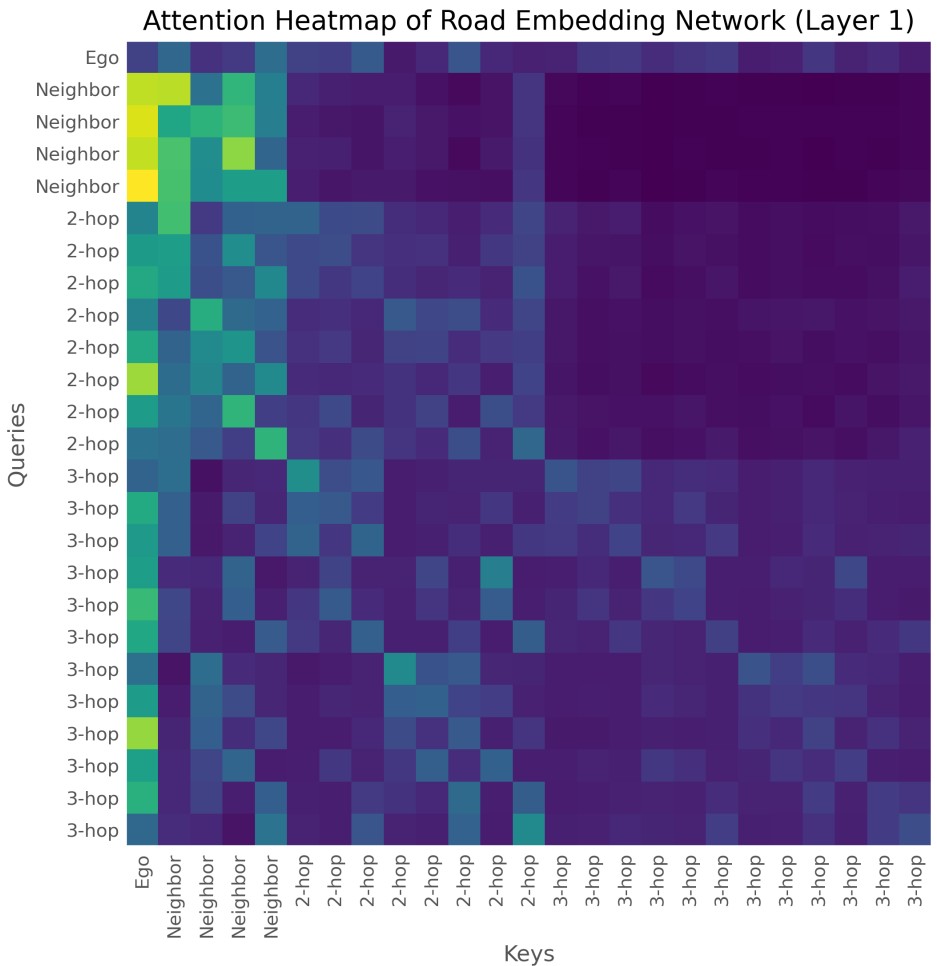

Figure 16: The attention weight matrix of the first transformer block in the road embedding subnetwork of a fully trained next-vertex prediction model. The model was given a node from a grid road network as input. Warmer colors indicate greater weight. "Ego" indicates the position encodes information about the input node, "neighbor" indicates encoded information about immediate neighbors of the input node, and "$k$-hop" indicates encoded information about $k$-hop neighbors of the input node.

## B.5    ABLATING THE ROAD EMBEDDING NETWORK

In this ablation experiment, plotted in Figure 17, we ablate the transformers from the road embedding network architecture. Recall that, to generalize our next-vertex prediction framework to routing on previously unseen subgraphs of large road networks, we began identifying road segments with features that consist of sequences of vectors. For every vertex in the road network, each vector in the vertex's feature sequence contains information about a particular $k$-hop neighbor of the vertex. Whereas our usual road embedding network architecture passes these sequences into transformers to obtain a flat embedding of each vertex/edge, here we remove the transformers and instead obtain embeddings by passing flattened feature sequences into multi-layer perceptron blocks.

**Experiment details**    This experiment is conducted on large grid road networks and our dataset of LLM-generated points-of-interest queries. We implement two networks in this experiment: one that implements a road embedding network consisting of a flattening operation and four MLP blocks, and another that implements a road embedding network consisting of four MLP blocks followed by four transformer blocks. All other parameters are kept constant and stated in full in Table 10.

Figure 17: Plot of next-vertex prediction accuracy for next-vertex prediction models with either a transformer-based road embedding network (Transformer) or a road embedding network that directly passes flattened inputs into multi-layer perceptrons (MLP).

**Observations.** The transformer blocks significantly improve train error and test error convergence compared to MLP. We attribute this to the fact that the transformer's inductive bias provides invariance about the ordering of the neighbors of a vertex. We note that the total numbers of free parameters with and without the transformer blocks are similar: while removing the transformer block removes free parameters, but removing invariance by flattening features into a vector also significantly increases the number of free parameters.

### B.6 ANALYZING END-TO-END LEARNED ROAD EMBEDDINGS

In Figure 18, we visualize the road embeddings learned by a fully trained next-vertex prediction model. In this figure, we compute the road embeddings for the vertices of 10 random small grid road networks and project them to a two-dimensional space by taking their t-distributed stochastic neighbor embeddings (TSNE) (Van der Maaten & Hinton, 2008). We then color the embedding points according to various semantic properties of their corresponding vertices. In particular, we color vertices by their latitude, or their longitude, or by their degree (number of immediate neighbors). We can visually confirm that vertices of a similar color indeed cluster together, rather than being scattered like random noise. This indicates that the road embedding network correctly learns to encode important geographic information—including a vertice's location within a grid road network—and important topological information—namely, a vertex's degree.

### B.7 HYPERPARAMETER TUNING

In this experiment, depicted in Figures 19 and 20, we perform hyperparameter tuning over the learning rate, weight decay, and learning rate schedules. We find learning rates between 0.0002 and 0.00005 to be most effective, and weight decay to be ineffective at any value. We similarly find negligible difference between implementing a cosine one-cycle learning rate and a linear one-cycle learning rate, at least for the short training timespan of these hyperparameter tuning runs. This experiment is conducted on large grid road networks and our dataset of simple template-generated points-of-interest queries. A single random seed is used for weight decay and learning rate tuning.

### B.8 VARYING BEAM WIDTH EXPERIMENT ON US ROAD NETWORKS

Figure 21 is an analogue of Figure 4 but on the US road network. In this experiment, we evaluate the same next-vertex prediction models from Table 2, again on previously unencountered queries

Figure 18: The t-distributed stochastic neighbor embeddings (TSNE) of the outputs of a fully-trained road embedding network given as input the vertices of 10 random small grid road networks. The embeddings are colored according to the degree of their corresponding vertices (top), the longitude of the vertices (middle), and the latitude of the vertices (bottom).

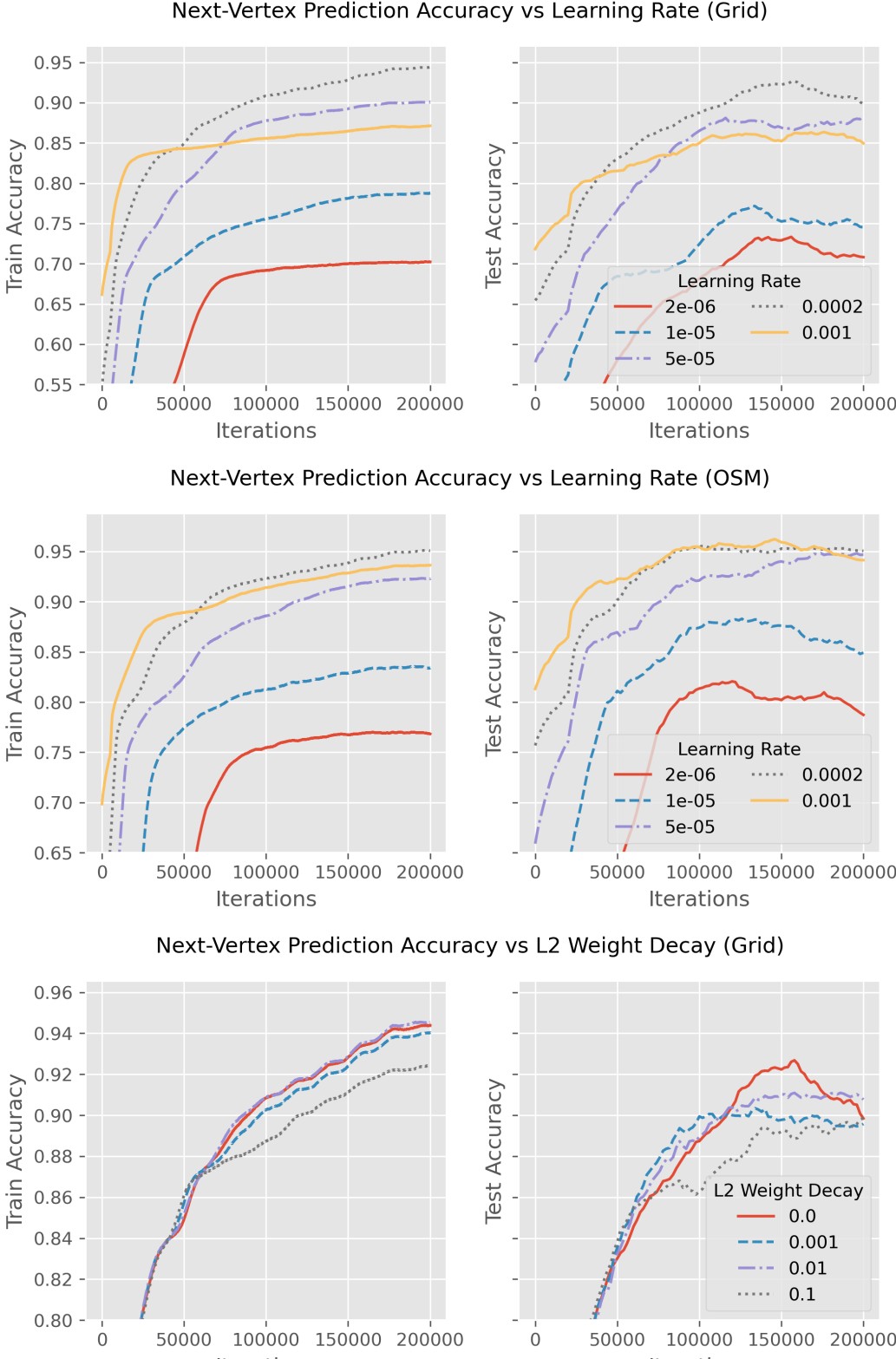

Figure 19: Plot of next-vertex prediction accuracy versus various choices of learning rates and L2 weight decay parameters. "OSM" indicates the experiment is conducted on real-world road networks in the United States. "Grid" indicates the experiment is conducted on simulated grid road networks.

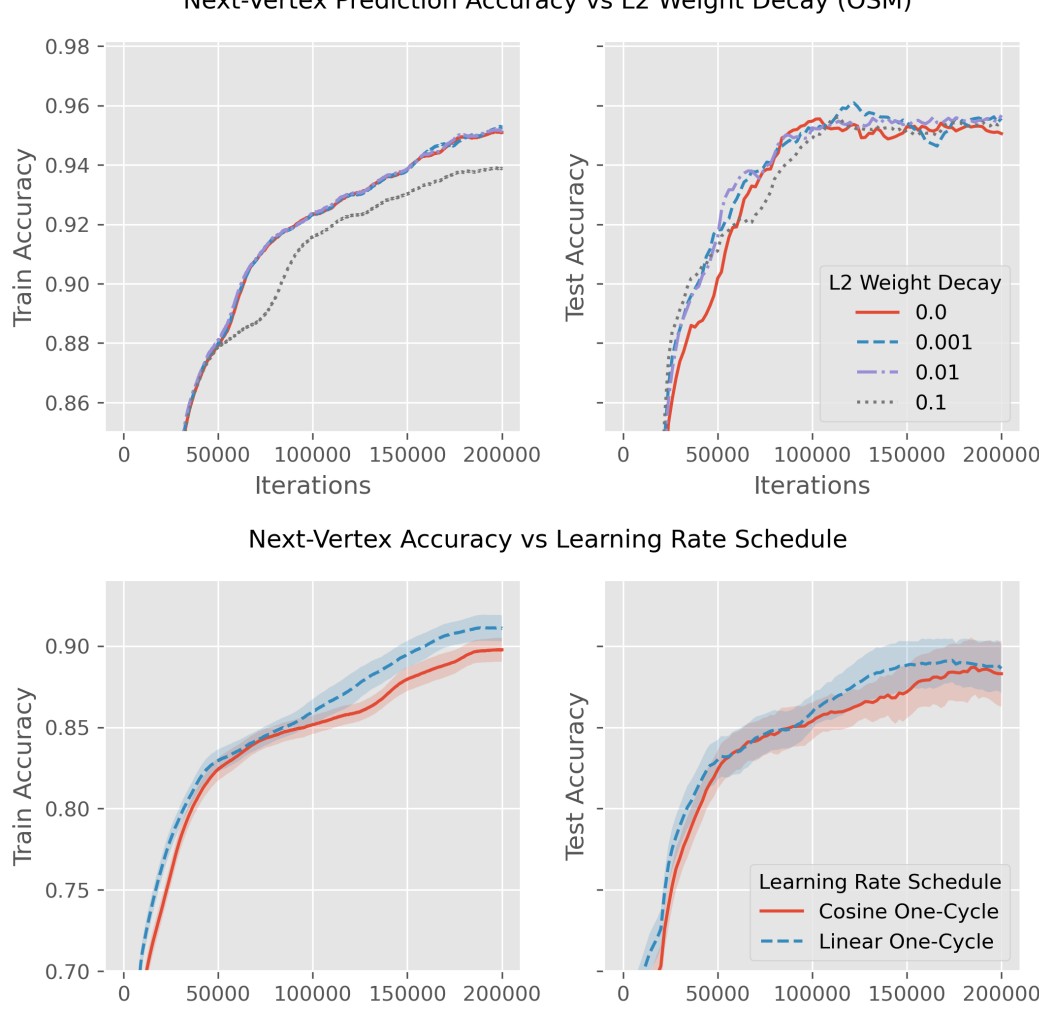

Figure 20: Plot of next-vertex prediction accuracy versus various choices of L2 weight decay parameters and learning rate schedules. "OSM" indicates the experiment is conducted on real-world road networks in the United States. "Grid" indicates the experiment is conducted on simulated grid road networks. Continues Figure 19.

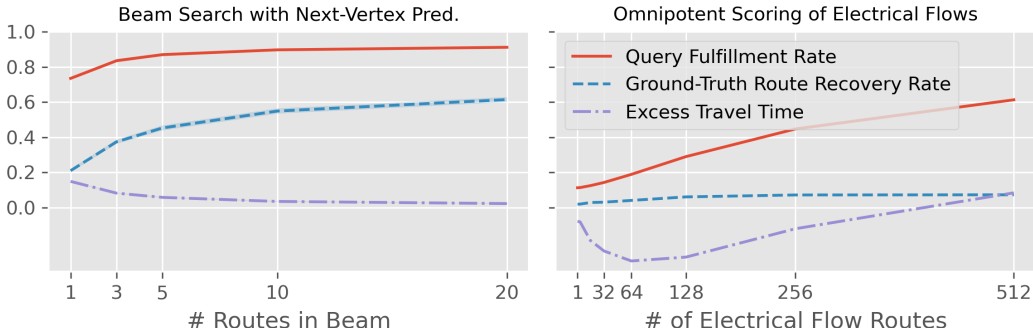

Figure 21: Performance on 256-vertex subgraphs on road networks from previously unseen US states and previously unseen roads queries. The left plot shows the performance of applying beam search with various widths. The right plot shows the performance of the best of $k$ candidate routes sampled with electrical flows for various choices of $k$.

and states in the US road network, but with varying beam widths. As with Figure 21, we observe monotonic improvements in all metrics with increasing beam-width, including improving upon the metrics reported in Table 2 by increasing beam width to 20. We note that the excess travel time metrics for the electrical flows baseline is sometimes negative; this is because the EF baseline often completely ignores user queries and instead takes the shortest route, allowing it to improve its travel times beyond what is achievable by a route that fulfills the user query.

## B.9 ABLATING THE SECONDARY SCORING MODEL

In this ablation experiment, plotted in Figure 22, we ablate the secondary scoring model from the customized routing experiments depicted in Tables 2 and 3. Recall that beam search with width $k$ ultimately produces up to $k$ candidate routes. In the experiments depicted in Tables 2 and 3, we trained a second network to predict the best route. In this experiment, we empirically compare against two alternatives. The first alternative follows the canonical beam search implementation and chooses a candidate based off the next-vertex prediction model's logits. The second alternative uses a ground-truth "omnipotent" scorer to choose the best route among the candidates, which prioritizes fulfilling the user query over minimizing travel time. We observe that using the secondary scoring model significantly improves routing performance, increasing most metrics to around that of the ground-truth scorer (the gold standard).

## B.10 INFERENCE ALGORITHMS

In this experiment, we study choices of inference algorithm for generating routes using next-vertex prediction models. We explore two alternates to the naive strategy of greedy decoding. The first is Beam Search, which is a standard algorithm in natural language generation and the primary inference algorithm in our experiments. The second is Dijkstra's algorithm—or more correctly, a generalization of Dijkstra's algorithm compatible with route objective functions that do not decompose additively into edge costs—which is a classical shortest-paths algorithm.

**Beam search.** Beam search with width $k$ maintains a set of $k$ partial routes $(v_1^1, v_2^1, \ldots, v_i^1), \ldots, (v_1^k, v_2^k, \ldots, v_i^k)$ that it rolls out simultaneously. At each time step, beam search explores all partial routes that can be formed by appending a node to one of its current $k$ routes, selecting its partial route set for the next time step by choosing the $k$ assigned the highest joint probability by the next-vertex prediction model.

**Dijkstra's algorithm.** We define a generalized form of Dijkstra's algorithm that accommodates a notion of "width" as follows. Given a width parameter of $k$, this algorithm maintains counters for each vertex in the graph and also a set of partial routes. The algorithm's partial route set is

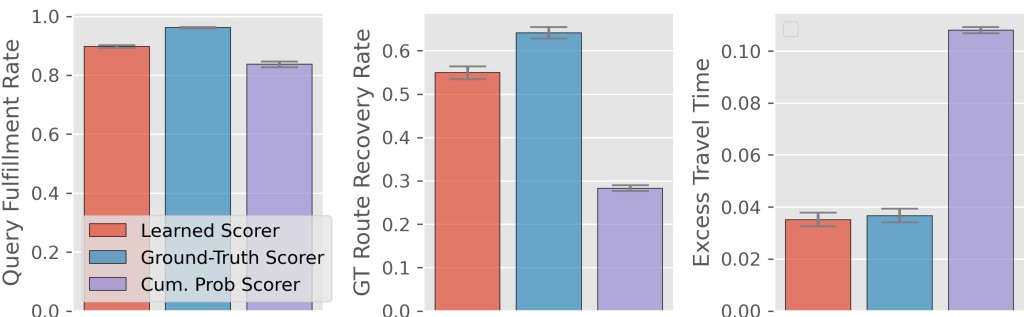

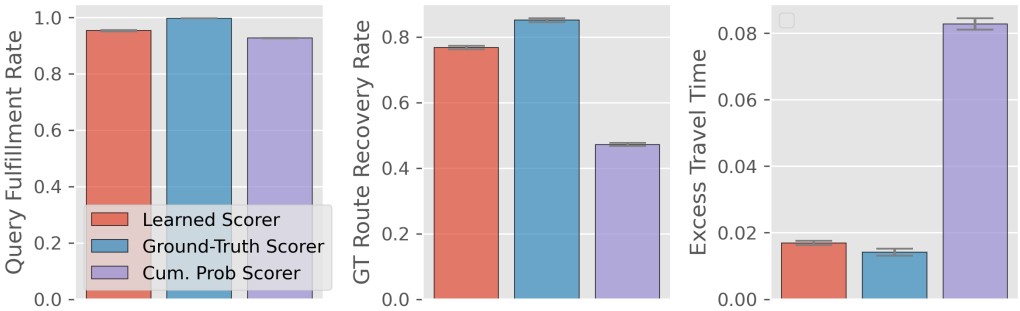

Figure 22: Bar plot of the performance metrics of using beam search on a next-vertex prediction model on previously unseen queries and roads from either the United States road network (top) or a large grid road network (bottom). "Learned scorer" refers to choosing from a route from the final beam using a secondary scoring model, while "Ground-Truth Scorer" refers to using an omnipotent scorer and "Cum. Prob Scorer" refers to scoring using the next-vertex prediction model's logits.

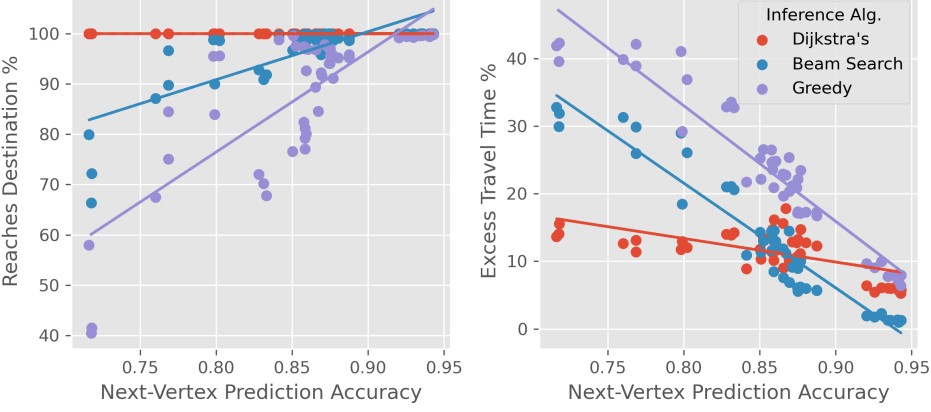

Figure 23: Point plot of the routing performance metrics of various next-vertex prediction models using either greedy decoding, beam search, or Dijkstra's algorithm as an inference method. These prediction models are all trained on some variant of a grid road network routing dataset and are evaluated on previously unseen customized routing tasks.

of unrestricted size and may consist of partial routes of different lengths. At each time step, the algorithm explores all partial routes that can be formed by appending a node to one of its current partial routes. The algorithm then adds the route $(v_1, \ldots, v_i, v_{i+1})$ that is assigned the highest joint probability to its partial route set and increments the counter for the vertex $v_i$. If the counter for a vertex reaches $k$, the algorithm can no longer consider expanding partial routes that end in $v_i$. Note that setting $k = 1$ would correspond to running the standard Dijkstra's algorithm where each vertex is expanded at most once.

**Observations.** We generally find that beam-search to be the most effective inference algorithm given a powerful next-vertex prediction model. However, we also find that Dijkstra's algorithm can be a more effective inference method than beam search when one's underlying next-vertex prediction model is inaccurate. We highlight this in Figure 23, where we evaluate a diverse set of next-vertex prediction models representing both successful and failed training runs. These models and their evaluation metrics are aggregated from many different experiment settings, but all defined on some grid road network navigation task. We can observe that, indeed, beam search significantly outperforms Dijkstra's algorithm as an inference method for highly accurate next-vertex prediction models. However, with models of lower accuracy, Dijkstra's algorithm serves as a robust alternative, providing significantly better rates for excess travel time and destination reaching than beam search on less performant next-vertex prediction models.

| Road Network Type | Vertices $n$ | Edges | POI $\lambda$ | Highways $m$ | Avg. Travel Time $\mu$ |
|---|---|---|---|---|---|
| Small Grid | 625 | 2462 | 0.5% | $n \mathbin{/\mkern-5mu/} 20$ | 5 |
| Large Grid | $\infty$ | $\infty$ | 0.5% | $\infty$ | 5 |
| Subgraph of Large Grid | 256 | 984 | 0.5% | $n \mathbin{/\mkern-5mu/} 20$ | 5 |

| Road Network Type | Vertices $n$ | Edges | POI $\lambda$ | Pruned Leaf Vertices |
|---|---|---|---|---|
| United States (OSM) | 70M+ | 100M+ | 0.05% | 80% |
| Subgraph of United States (OSM) | 256 | 984 | 0.05% | 80% |

Table 4: The size and parameters for our experiments' road networks. POI $\lambda$ describes the probability that a given vertex is a point-of-interest of a given category. Avg. Travel Time describes the mean of the Poisson distribution from which road travel times are sampled. Pruned Leaf Vertices describes the percentage of leaf vertices pruned from the road networks provided by OpenStreetMaps.

## C    EXPERIMENT DETAILS

### C.1    ROAD NETWORKS

We experiments on two categories of road networks: simulated grid road networks and real-world road networks from the OpenStreetMap repository (OpenStreetMap contributors, 2017).

1. **Grid road networks from simulation.** The grid road networks are randomly generated and designed to approximate the topology of real-world road networks. We construct these networks by initializing a regular grid of $n$ vertices with $\sqrt{n}$ rows and $\sqrt{n}$ columns and bidirectional edges drawn between every vertex and its cardinal neighbors. The travel times for each road segment—i.e., the amount of time it takes to transit across an edge in the road network—are assigned randomly by sampling from a Poisson distribution with a parameter $\mu$. Additional "highway" roads are then added between $m$ randomly chosen pairs of vertices in the grid that are at least 3 hops apart; all such highway roads are assigned a travel time of 1, regardless of length. We then randomly designate vertices points-of-interest (POI) for various categories of POIs, such as coffee shops, with probability $\lambda$.

2. **Real-world road networks of the United States.** We also study the real-world road networks of various states and territories in the United States, which we obtain through the open-source map data provided by the OpenStreetMap project (OpenStreetMap contributors, 2017). We lightly preprocess these road networks to ensure that routing tasks on these networks are non-trivial. First, we contract vertices of degree two from the road networks as navigating on these vertices is trivial (its a straight road). Second, we remove four out of every five leaf vertices from the road network—these vertices also contribute nothing to the difficulty of the routing tasks. To generate routing tasks from these road networks, we randomly select subgraphs by choosing a random point in the road network of a random state/territory, and take its $n$ closest neighbors. We assign a travel time to each road proportional to its real-world length, and randomly designate vertices to have points-of-interest with probability $\lambda$.

Table 4 specifies the parameter values used to preprocess these road networks.

**Points-of-interest.** Our experiments consider 42 categories of points-of-interest: coffee shops, gas stations, grocery stores, work offices, cat shelters, dinner restaurants, pharmacies, parks, museums, beaches, libraries, malls, fast food restaurants, post offices, car washes, bakeries, gyms, hardware stores, zoos, campgrounds, theaters, flower shops, bars, thrift stores, lakes, bridges, hospitals, churches, airports, banks, stadiums, police stations, spas, hotels, casinos, train stations, clubs, planetariums, tattoo parlors, internet cafes, bowling alleys, and ice cream shops.

| Dataset | User Query | Ground-Truth Interpretation |
|---|---|---|
| LLM-Generated Dataset | Go by pub. | "bar" |
| | Route through bank and feline center, and jog a lap. | "bank", "cat shelter", "park" |
| | Looking to grab a bite, enjoy a picnic, and relax. | "dinner", "lake", "spa" |
| | Gotta buy toiletries and either route through arena or home repair. | "pharmacy", either("stadium", "hardware store") |
| | Swing by postal service and airport, and either stop by vehicle wash or home improvement store. | "post office", "airport", either("car wash", "hardware store") |
| | Make a stop at fitness center and either route through parcel office or cultural visit. | "gym", either("post office", "museum") |
| Template-Generated Dataset | bar | "bar" |
| | [No Request] | |
| | dinner and flower shop | "dinner", "flower shop" |
| | internet cafe and bakery | "internet cafe", "bakery" |
| | work office and tattoo parlor | "work office", "tattoo parlor" |
| | bank and cat shelter and park | "bank", "cat shelter", "park" |

Figure 24: Randomly selected point-of-interest queries and their ground-truth interpretations from the LLM-generated dataset of ≈900M queries and the sentence-template-generated dataset of ≈80k queries.

**Train-test splits.** To measure generalization, we define a notion of a train-test split on each road network.

1. **Small grid road network.** We divide the set of all source-destination pairs on the small grid road network into training (95%) and testing (5%) splits. This means that our test set figures are reported for customized routing tasks that involve previously unencountered pairs of sources and destinations. This does not mean that, for every source-destination pair $u, v$ in the test split, the source vertex $u$ does not appear in the train split, as the train split may include routing tasks that involve routing from $u$ to a different destination $w \neq v$.

2. **Large grid road network.** We sample a set of square subgraphs of the large grid road network, and divide the set into training and testing splits. This means that our test set figures are reported for customized routing tasks that involve entirely unencountered subgraphs. This means that there is no notion of encountering the same vertex or edge in both the train set and the test set.

3. **United States road network.** We divide the states and territories of the United States into a test set containing Rhode Island, New Hampshire, and Mississippi and a train set containing the remaining 48 of 52 states and territories. The test set is chosen arbitrarily for their names (road island, new hamster, etc.). This means, as with the large grid road network, one will not encounter the same vertex and edge in both the train set and the test set.

## C.2 Natural Language Query Datasets

This paper performs experiments on two datasets of natural language queries. Semantically, the queries in these two datasets are similar. The queries in both of these datasets concern points-of-interest. That is, these queries request to navigate to a destination with the shortest travel time possible but subject to the constraint of needing to stop by certain (and often multiple) points of interest—such as a coffee shop and grocery store—along the way. These queries are often complex, and involve both AND conjunctions (stop by a coffee shop AND a grocery store on the way) and OR conjunctions (stop by a coffee shop OR a grocery store on the way home). Table 24 lists randomly sampled (not cherry-picked) examples of queries from each dataset and their ground-truth interpretation.

**Template-generated versus LLM-generated queries.** Our two datasets of example user queries differ in their syntax. The text queries in the first dataset are constructed according to a simple sentence template, such as "POI 1 and POI 2 and either POI 3 and POI 4". This yields a total of 79,927 unique queries in the first dataset. The text queries in the second dataset are constructed with the aide of the commercial large-language-model (LLM) GPT-4 (OpenAI, 2023). Specifically, we asked the GPT-4 LLM to generate a corpus of phrases and synonyms associated with each point-of-interest, such as associating "grabbing an espresso" with "coffee shops". We further asked the LLM to produce a corpus of sentence structures that a user query may follow, such as "Route through {} and {}", substituting in the LLM's recommendations for POI-associated phrases accordingly. This yields a total of 913,453,361 unique queries in the second dataset.

**Train-test splits.** We divide the queries in each query dataset into a train split and test split. We also note that we want to validate the generalization of our models to queries of not only previously unencountered syntax but also of previously unencountered semantics. To this end, we perform a train-test split of each query dataset by splitting the queries according to their semantic meaning (that is, their ground-truth interpretation). This means that if a customized routing task from the test set asks to "stop by a coffee shop or grocery store", the train set will neither contain this same query nor any rephrasing of the query, e.g. "either grab an espresso or grab groceries".

**Semantic complexity.** Figure 25 plots a histogram of the number of logical clauses in each of the queries in the natural language datasets. For example, the query "visit the library and either grab an espresso or go to the grocery store" consists of 3 clauses. Note that the queries in the test split of the datasets are significantly more complex and challenging than the queries in the train split.

## C.3 Putting Road Networks and Query Datasets Together

Recall that every datapoint in our training sets of customized routing tasks consists of three parts: a natural language query, a source and destination pair, and a ground-truth route. Figure 26 plots two examples, one from a grid road network and another from the United States road network.

**Train split.** We construct a training dataset of $N$ customized routing tasks by sampling $N$ source-destination pairs from the train split of the road network and sampling $N$ queries from the train split of the query dataset. Recall that our query dataset also provides labels for these $N$ queries in the form of structured ground-truth interpretations. We therefore obtain the ground-truth routes for our $N$ datapoints by applying brute-force traveling salesman problem (TSP) algorithms to the POI routing problems specified by the ground-truth interpretations of the $N$ queries on their respective source-destination pairs.

**Test split.** Similarly, we construct the testing dataset of $M$ customized routing tasks by sampling $M$ datapoints from the test split of the road network and $M$ queries from the test split of the query dataset. In Tables 2 and 3 we also evaluate on a secondary testing dataset consisting of $M$ datapoints from the test split of the road network and $M$ queries from *train* split of the query dataset.

**Statistics.** Table 5 lists important statistics for the customized routing tasks datasets that we generate with this approach.

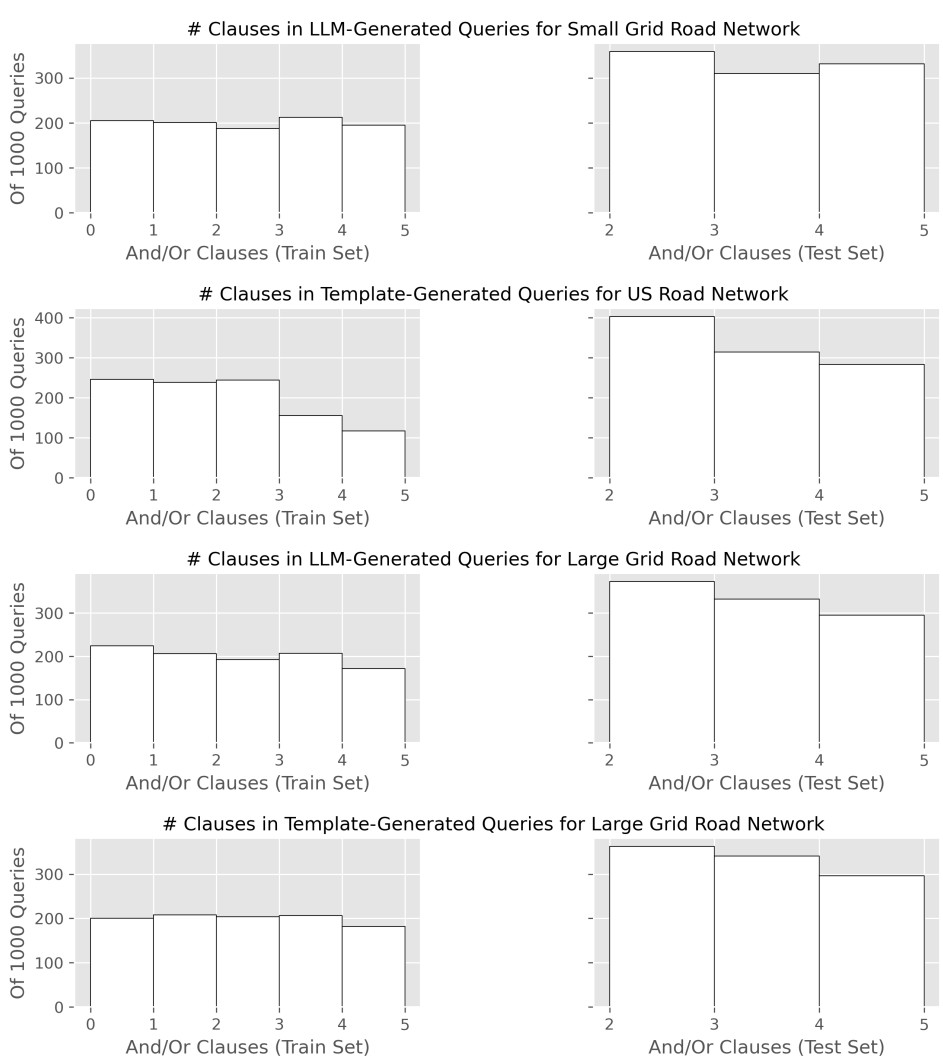

Figure 25: Histogram of the number of clauses in natural language queries for each dataset.

"Swing by beauty salon and pick up a magazine"

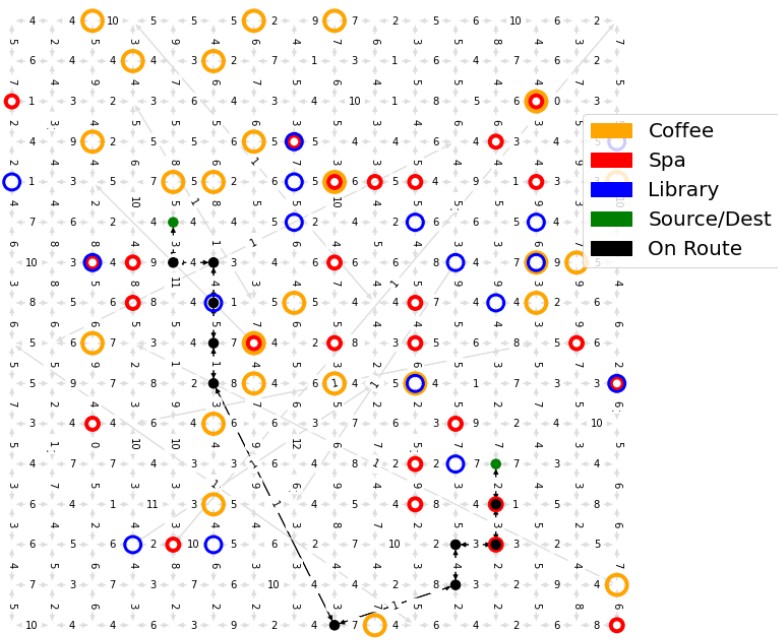

"Need to get some reading and get a mocha"

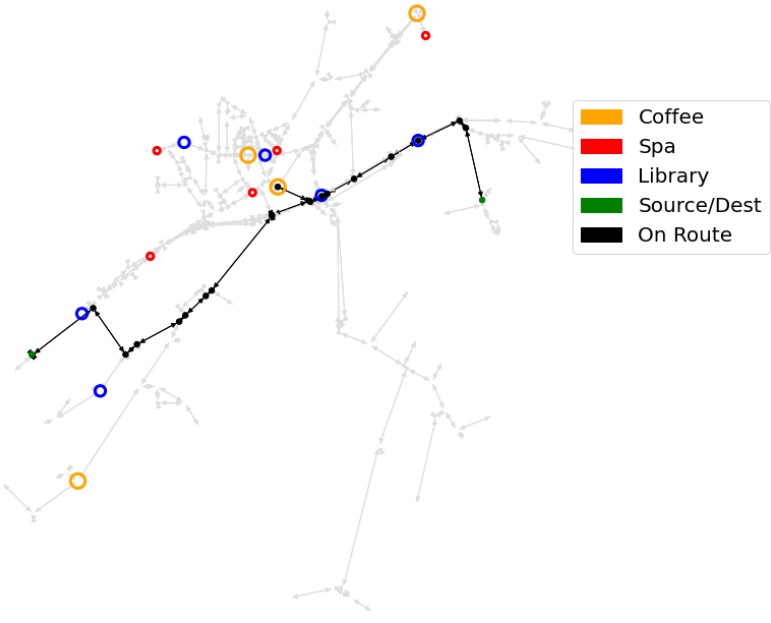

Figure 26: Examples of customized routing tasks from a 256 vertex subgraph of a simulated grid road network and a 256 vertex subgraph of the United States road network. For ease of visualization, the road networks are simplified to only include three (rather than the usual 42) point-of-interest categories. The ground-truth route is depicted as a black line, with the source and destination vertices colored green. Points-of-interest are indicated with colored circles, with concentric circles indicating a location has multiple points-of-interest.

| | Metric | Train Set | Test Set |
|---|---|---|---|
| United States Road Network | # Segments in Ground-Truth Route | 19.1 | 21.3 |
| | Diameter of Ground-Truth Route | 14.6 | 14.8 |
| | # of Clauses in Query | 1.9 | 2.9 |
| | # Segments in Shortest Route | 13.8 | 13.5 |
| Large Grid Road Network | # Segments in Ground-Truth Route | 10.0 | 10.9 |
| | Diameter of Ground-Truth Route | 8.8 | 9.0 |
| | # of Clauses in Query | 1.9 | 3.0 |
| | # Segments in Shortest Route | 8.4 | 8.4 |
| Small Grid Road Network | # Segments in Ground-Truth Route | 11.1 | 11.7 |
| | Diameter of Ground-Truth Route | 10.2 | 10.2 |
| | # of Clauses in Query | 1.8 | 3.0 |
| | # Segments in Shortest Route | 9.9 | 9.8 |

Table 5: Customized routing task dataset statistics estimated with 1,000 samples from each split.

### C.4 HYPERPARAMETERS

In this section, we detail the hyperparameters for each experiment in this paper. We first clarify terminology for the below tables.

**Network architecture.** We use "Embedding Dim. (Base)", "Intermediate Dim. (Base)", and "Attention Heads (Base)" to denote the embedding dimension, intermediate dimension, and attention head count of the transformer blocks in the base network. Similarly, we use "Embedding Dim. (Road)" and "Intermediate Dim. (Road)" denote the embedding and intermediate dimensions of the transformer blocks and MLP blocks in the road embedding network. "Attention Heads (Road Emb)" denotes the attention head count of the transformer blocks in the road embedding network, while "Addtl. MLP Blocks (Road Emb)" denotes the number of MLP blocks placed before the transformer blocks in the road embedding network.

**Road features.** The "Road Feature Type" key refers to what type of features we associate to each vertex/edge when producing an embedding for each vertex/edge. That is, it refers to the input format of the road embedding network. A "Flat Vector" road feature type denotes when each vertex's features are flattened into a 1d vector so it can be projected into a 1d embedding. A "Seq. Vectors" road feature type denotes when a vertex's features are passed as a 2d matrix, corresponding to a sequence of feature vectors, which must then be collapsed into a 1d embedding. The "Road Feat. Receptive Field" key refers to the receptive field encoded in the features we associate to each vertex. When the value is "1-hop", this means that the road embedding network can only take into account the properties of a node $v$ and its immediate neighbors when producing an embedding of $v$. A value of "3-hop" means that the road embedding network can instead take into account the 3-hop neighbors of $v$ when producing an embedding for $v$.

**Miscellaneous.** In the "Learning Rate Schedule" key, "Linear OC" denotes that a linear one-cycle learning rate schedule is used with default shape parameters while "Cosine OC" denotes that a cosine

| Road Feat. Receptive Field | 1-hop | 2-hop | 3-hop | 4-hop |
|---|---|---|---|---|
| Datapoints | 20000000 | 20000000 | 20000000 | 20000000 |
| Learning Rate | 0.0001 | 0.0001 | 0.0001 | 0.0001 |
| L2 Weight Decay | 0 | 0 | 0 | 0 |
| Dropout | 0 | 0 | 0 | 0 |
| Batch Size | 1024 | 1024 | 1024 | 1024 |
| Training Iterations | 200000 | 200000 | 200000 | 200000 |
| Seeds | 3 | 3 | 3 | 3 |
| Learning Rate Schedule | Linear OC | Linear OC | Linear OC | Linear OC |
| Attention Heads (Base) | 8 | 8 | 8 | 8 |
| Embedding Dim. (Base) | 512 | 512 | 512 | 512 |
| Intermediate Dim. (Base) | 1024 | 1024 | 1024 | 1024 |
| Transformer Blocks (Base) | 8 | 8 | 8 | 8 |
| Road Feature Type | Seq. Vectors | Seq. Vectors | Seq. Vectors | Seq. Vectors |
| Attention Heads (Road) | 4 | 4 | 4 | 4 |
| Embedding Dim. (Road) | 128 | 128 | 128 | 128 |
| Intermediate Dim. (Road) | 256 | 256 | 256 | 256 |
| Transformer Blocks (Road) | 4 | 4 | 4 | 4 |
| Addtl. MLP Blocks (Road) | 4 | 4 | 4 | 4 |

Table 6: Hyperparameters for the road features receptive field experiment (Appendix B.1).

| POI Density | 0.005 | 0.0005 | 0.05 |
|---|---|---|---|
| Datapoints | 20,000,000 | 20,000,000 | 20,000,000 |
| Learning Rate | 0.0001 | 0.0001 | 0.0001 |
| L2 Weight Decay | 0 | 0 | 0 |
| Dropout | 0 | 0 | 0 |
| Batch Size | 1024 | 1024 | 1024 |
| Training Iterations | 200,000 | 200,000 | 200,000 |
| Seeds | 3 | 3 | 3 |
| Learning Rate Schedule | Linear OC | Linear OC | Linear OC |
| Road Feat. Receptive Field | 3-hop | 3-hop | 3-hop |
| Attention Heads (Base) | 8 | 8 | 8 |
| Embedding Dim. (Base) | 512 | 512 | 512 |
| Intermediate Dim. (Base) | 1024 | 1024 | 1024 |
| Transformer Blocks (Base) | 8 | 8 | 8 |
| Road Feature Type | Seq. Vectors | Seq. Vectors | Seq. Vectors |
| Attention Heads (Road) | 4 | 4 | 4 |
| Embedding Dim. (Road) | 128 | 128 | 128 |
| Intermediate Dim. (Road) | 256 | 256 | 256 |
| Transformer Blocks (Road) | 4 | 4 | 4 |
| Addtl. MLP Blocks (Road) | 4 | 4 | 4 |

Table 7: Hyperparameters for the point-of-interest density experiment (Appendix B.3), which is the only experiment for which the POI Density parameter deviates from the default values specified in Table 4.

one-cycle learning rate schedule is used instead. The "Seeds" key refers to the number of random seeds the experiment is performed on.

| Attention Heads (Road) | 1 | 2 | 4 | 4 |
|---|---|---|---|---|
| Addtl. MLP Blocks (Road) | 1 | 2 | 2 | 4 |
| Transformer Blocks (Road) | 1 | 2 | 4 | 4 |
| Datapoints | 20,000,000 | 20,000,000 | 20,000,000 | 20,000,000 |
| Learning Rate | 0.0001 | 0.0001 | 0.0001 | 0.0001 |
| L2 Weight Decay | 0 | 0 | 0 | 0 |
| Dropout | 0 | 0 | 0 | 0 |
| Batch Size | 1024 | 1024 | 1024 | 1024 |
| Training Iterations | 200,000 | 200,000 | 200,000 | 200,000 |
| Seeds | 3 | 3 | 3 | 3 |
| Learning Rate Schedule | Linear OC | Linear OC | Linear OC | Linear OC |
| Road Feat. Receptive Field | 3-hop | 3-hop | 3-hop | 3-hop |
| Attention Heads (Base) | 8 | 8 | 8 | 8 |
| Embedding Dim. (Base) | 512 | 512 | 512 | 512 |
| Intermediate Dim. (Base) | 1024 | 1024 | 1024 | 1024 |
| Transformer Blocks (Base) | 8 | 8 | 8 | 8 |
| Road Feature Type | Seq. Vectors | Seq. Vectors | Seq. Vectors | Seq. Vectors |
| Embedding Dim. (Road) | 128 | 128 | 128 | 128 |
| Intermediate Dim. (Road) | 256 | 256 | 256 | 256 |

Table 8: Hyperparameters for the road embedding network scaling experiment (Appendix B.2).

| Attention Heads (Base) | 8 | 4 | 4 | 8 |
|---|---|---|---|---|
| Embedding Dim. (Base) | 512 | 64 | 256 | 1024 |
| Intermediate Dim. (Base) | 1024 | 256 | 512 | 2048 |
| Transformer Blocks (Base) | 8 | 4 | 6 | 8 |
| Embedding Dim. (Road) | 128 | 64 | 64 | 128 |
| Intermediate Dim. (Road) | 256 | 128 | 256 | 512 |
| Datapoints | 20,000,000 | 20,000,000 | 20,000,000 | 20,000,000 |
| Learning Rate | 0.0001 | 0.0001 | 0.0001 | 0.0001 |
| L2 Weight Decay | 0 | 0 | 0 | 0 |
| Dropout | 0 | 0 | 0 | 0 |
| Batch Size | 1024 | 1024 | 1024 | 1024 |
| Training Iterations | 200,000 | 200,000 | 200,000 | 200,000 |
| Seeds | 3 | 3 | 3 | 3 |
| Learning Rate Schedule | Linear OC | Linear OC | Linear OC | Linear OC |
| Road Feat. Receptive Field | 3-hop | 3-hop | 3-hop | 3-hop |
| Road Feature Type | Seq. Vectors | Seq. Vectors | Seq. Vectors | Seq. Vectors |
| Attention Heads (Road) | 4 | 4 | 4 | 4 |
| Addtl. MLP Blocks (Road) | 4 | 4 | 4 | 4 |
| Transformer Blocks (Road) | 4 | 4 | 4 | 4 |

Table 9: Hyperparameters for the embedding dimension scaling experiment (Appendix B.2).

| Road Feature Type | Flat Vector | Seq. Vectors |
|---|---|---|
| Datapoints | 20,000,000 | 20,000,000 |
| Learning Rate | 0.0001 | 0.0001 |
| L2 Weight Decay | 0 | 0 |
| Dropout | 0 | 0 |
| Batch Size | 1024 | 1024 |
| Training Iterations | 200,000 | 200,000 |
| Seeds | 3 | 3 |
| Learning Rate Schedule | Linear OC | Linear OC |
| Road Feat. Receptive Field | 3-hop | 3-hop |
| Attention Heads (Base) | 8 | 8 |
| Embedding Dim. (Base) | 512 | 512 |
| Intermediate Dim. (Base) | 1024 | 1024 |
| Transformer Blocks (Base) | 8 | 8 |
| Attention Heads (Road) | N/A | 4 |
| Embedding Dim. (Road) | 128 | 128 |
| Intermediate Dim. (Road) | 256 | 256 |
| Addtl. MLP Blocks (Road) | 4 | 4 |
| Transformer Blocks (Road) | N/A | 4 |

Table 10: Hyperparameters for the road embedding architecture experiment (Appendix B.5).

| Road Network | United States | Large Grid | Small Grid |
|---|---|---|---|
| Datapoints | 100,000,000 | 100,000,000 | 20,000,000 |
| Learning Rate | 0.0001 | 0.0001 | 0.0001 |
| L2 Weight Decay | 0 | 0 | 0 |
| Dropout | - | - | - |
| Batch Size | 1024 | 1024 | 1024 |
| Training Iterations | 1,000,000 | 1,000,000 | 200,000 |
| Seeds | 5 | 5 | 5 |
| Learning Rate Schedule | Cosine OC | Cosine OC | Cosine OC |
| Road Feat. Receptive Field | 3-hop | 3-hop | N/A |
| Attention Heads (Base) | 8 | 8 | 8 |
| Embedding Dim. (Base) | 1024 | 1024 | 512 |
| Intermediate Dim. (Base) | 2048 | 2048 | 1024 |
| Transformer Blocks (Base) | 8 | 8 | 8 |
| Road Feature Type | Seq. Vectors | Seq. Vectors | IDs |
| Attention Heads (Road) | 2 | 2 | N/A |
| Embedding Dim. (Road) | 128 | 256 | N/A |
| Intermediate Dim. (Road) | 512 | 512 | N/A |
| Addtl. MLP Blocks (Road) | 2 | 2 | N/A |
| Transformer Blocks (Road) | 2 | 2 | N/A |

Table 11: Hyperparameters for the primary experiments of the paper (Tables 1, 2, 3).

| | |
|---|---|
| Datapoints | 125,000 |
| Learning Rate | 0.0005 |
| L2 Weight Decay | 0.01 |
| Batch Size | 128 |
| Training Iterations | 20,000 |
| Intermediate Dim. | 64 |
| Hidden Layers | 1 |
| Activation | GELU |

Table 12: Hyperparameters for the secondary scoring model in all experiments (Tables 3, 2).

