# OpenReview forum: "Routing with Rich Text Queries via Next-Vertex Prediction Models"
_ICLR.cc/2024/Conference — Submitted to ICLR 2024_

### Official Review · Reviewer_P8sR · 2023-10-18

**Soundness:** 3 good
**Presentation:** 3 good
**Contribution:** 2 fair
**Rating:** 5
**Confidence:** 3

**Summary:**

This paper considers the problem of predicting shortest paths in networks under a variety of possible constraints which are specified in natural language. The authors present a language modeling inspired approach to predicting shortest paths by using a transformer to predict the next vertex in path given a partial path. The model jointly encodes the natural language constraints, the source vertex, the destination vertex, the path taken so far, and candidates for the next vertex and predicts the next vertex in the path. The model is trained autoregressively on ground truth paths and beam search decoding is used to predict the shortest path at inference time. Experimental results are show that the trained model is able to produce high-quality paths given natural language queries on real-world datasets.

**Strengths:**

* The paper is very well written and the figures are clear and helpful
* The proposed approach is elegant and effectively applies progress in language modeling to the problem of constrained routing
* The results show that the method is effective in producing high-quality paths under the natural language constraints

**Weaknesses:**

* There is a lack of comparable baselines to the proposed method. The authors claim that the unguided search electrical flows with omnipotent referee is a strong baseline, but several other methods are mentioned in the introduction and related work seem to be viable candidates. If these methods are inadequate as the authors claim, then results should be shown against a reasonable cross section of such methods showing where these methods fail and the proposed method succeeds.
* The results do not elucidate why the proposed method works. The authors present a large volume of great results, but none of which that explain how the model is working. It seems like there is a lack of information for the model to adequately plan a route using just local information in the network.
* The complexity of the natural language queries does not seem to be a large component of the solution, but does seem to be a large component of how complex the problem could be in practice. It seems like the method is just converting natural language to a one-hot encoding a points of interest and matching those with the points of interest in the receptive field of the encoded vertices in the input. Are the natural language queries in the experimental results accomplishing more than just one-hot matching between the natural language query and the points of interest in the receptive field?

**Questions:**

* Does the performance of the model change as the routes become longer?
* Are there particular types of instances where the model performs quite well and other instances where the model performs poorly?
* Based on Figure 11, it appears that the method does not benefit from a larger than a 2 (or 3)-hop receptive field. Is this a fact of the types of routes being considered at training and test time? Could the method benefit from a larger receptive field if the
* What information is the model using to "plan" its route? Is it using the coordinates of the destination relative to the source, and just choosing next vertices which greedily move in that direction (analogously for points of interest specified in the constraints)? It seems like there should not always be enough information in the input to accurately predict the next vertex on the path? Are there any examples of failure cases of this method?

---

> ### Author Response · Authors · 2023-11-16
> **Response to Reviewer P8sR**
>
> We appreciate the reviewer’s thoughtful feedback. We address your main questions below.
>
> **Complexity of natural language queries**
>
> Our method does not use structured query inputs that can be directly mapped to one-hot encodings. Rather, it directly takes as input unstructured natural language queries from users. Our dataset of natural language user queries are complex and diverse (see Table 24), and involve “and”/”or” combinations that require explicit reasoning.
> We can be confident that our models perform some form of non-trivial natural language processing since we use a test set that exclusively includes previously unseen user queries.
>
> **Why does the method work despite using only local information? Are there instances where the model performs poorly? Why does the model not benefit from a larger receptive field?**
>
> We agree it may be initially surprising that only local information is needed for constrained routing. Two factors play a role.
>
> 1) *Real-world road networks (which is what our experiments use) are easier to route on than worst-case graphs.* This is because travel time between locations usually approximates geographic distance up to small constant factors. Our model performs poorly on graphs that deviate from this, e.g. where the fastest route to a destination 1km away is to route through a midpoint 50km away.
>
> 2) *In beam search, each candidate route allows our model to explore a different local region of the graph.* Our models may therefore be limited to using local information but can be strategic about where to collect this local information. In fact, we can think of beam search as performing some efficient graph search. Even with a 1-hop receptive field, a sufficiently large beam width lets one observe the entire graph before choosing a route. Our model not benefiting from a larger receptive field (Figure 11) is also therefore because the most effective way to give our model more global information is to increase its beam width (Figure 4).
>
> **Comparisons**
>
> The main contribution of our paper is demonstrating that a next-vertex prediction approach yields surprisingly good results for natural language constrained routing. To the best of our knowledge, there are only two alternative approaches: the use of cost modifiers (learning edge costs so one can run an algorithm like Dijkstra’s) and the use of unguided search algorithms (e.g., using electrical flows). For the former, it is immediate that it is impossible for cost modifiers to handle point-of-interest-like queries, which would require route-level costs that cannot be decomposed additively into edge costs. For the latter, we do include an electrical flows baseline, which clearly demonstrates that the routing problems in our experiment are non-trivial and difficult. We are open to implementing other baseline alternatives to our next-vertex prediction approach, but are not aware of any others.
>
> We also want to note that there are ways that one could have implemented our next-vertex prediction approach differently. One could have used a different network to embed the receptive field, e.g. with a graph neural network rather than a transformer. However, we believe that we already have found a reasonable and effective implementation that demonstrates the efficacy of a next-vertex prediction approach; further architectural tunings are interesting but beyond the scope of this paper.
>
> **Does the performance of the model change as the routes become longer?**
>
> We evaluated our models on routing problems that require, on average, >21 non-trivial instructions, e.g., turn left (Table 5). This is among the complex end of problems that one would encounter in real-world applications. We did observe some degradation in performance with longer routes, but found the degradation disappears as one scales up one’s model. We will add a section to the Appendix with said result.
>
> **Does the model observe coordinates during planning?**
>
> Our model is able to observe the coordinates of the destination, source, and every vertex within a k-hop neighborhood of one’s candidate route.

---

> > ### Comment · Reviewer_P8sR · 2023-11-21
> >
> > Thank you for your response. I still believe that other baselines (maybe not existing in the literature explicitly) should be included and compared to in the paper. For example, one idea could be to train a model which predicts the cost modifiers on the network given the natural language query. This might not work very well, but it is at least some other baseline. The electrical flows pseudo-oracle method seems like too much of a straw man even with the provided oracle information.
> >
> > That being said, this paper has a lot of promise, but in my opinion is not quite ready for publication.

---

### Official Review · Reviewer_eXjd · 2023-10-31

**Soundness:** 3 good
**Presentation:** 2 fair
**Contribution:** 2 fair
**Rating:** 6
**Confidence:** 2

**Summary:**

This paper presents a novel approach to routing problems on graphs using autoregressive modeling and transformers. The authors propose a multimodal architecture that jointly encodes text and graph data and trains it via next token prediction. They demonstrate the effectiveness of their approach on synthetic graphs and real-world data from the OpenStreetMap repository.

**Strengths:**

- The proposed method is innovative and practical, effectively combining autoregressive modeling and transformers for routing problems. This approach allows the model to process complex queries in natural language and produce near-optimal routes, which is a significant improvement over traditional combinatorial methods.

- The paper provides extensive experiments on both synthetic and real-world data, showcasing the effectiveness of the approach. The results demonstrate that the proposed method can achieve high accuracy and efficiency in routing tasks, even when dealing with large-scale graphs and complex queries.

**Weaknesses:**

- The paper could benefit from more detailed implementation details to facilitate the reproducibility of the study. Providing code or more information on the specific choices of model architecture, training procedures, and evaluation metrics would make it easier for other researchers to replicate and build upon the proposed method.


- The authors could include more ablation studies to better understand the impact of different components of the proposed method, detailed analysis could provide a more comprehensive understanding of the factors that contribute to the method's success.

**Questions:**

- Could you provide codes or more information on the specific implementation details of the proposed method, such as the model architecture, hyperparameters, and training procedures? This would help other researchers to better understand and reproduce the proposed method.

- How does the proposed method compare to other state-of-the-art approaches in terms of computational efficiency and scalability? Providing a detailed analysis of the method's performance on large-scale graphs and complex queries would help to establish its advantages over existing techniques.

- Are there any potential limitations or challenges in applying the proposed method to real-world scenarios with more complex constraints and larger datasets? Discussing these limitations and potential solutions would provide a more comprehensive understanding of the method's applicability and future research directions.

---

> ### Author Response · Authors · 2023-11-16
> **Response to Reviewer eXjd**
>
> We thank the reviewer for their response. We address your questions below.
>
> **Replication**
>
> Experiment details for reproducibility are already provided in Section C. This includes dataset details and dataset statistics, details on the choice of model architecture (C.1-C.3) and model hyperparameters (C.4). Hyperparameter tuning details are also already provided in Section B.7. We will be releasing the full source code for these experiments in the de-anonymized version of this paper post-acceptance.
>
> **Ablation studies**
>
> We have conducted extensive ablation studies. As noted on the bottom of Page 9, we include in the Appendix additional experiments that study the relationship between model performance and (1) how much of the road network the models observe (Figure 11), (2) the scale of the models’ network architectures (Figure 12, 13), (3) the difficulty of a dataset’s route customization queries (Figure 14), and (4) one’s choice of inference algorithm (Figure 23). We also perform ablation studies of the road embedding network (Figure 17) and secondary scoring model (Figure 22). We also visualize and analyze the attention matrices of our networks (Figures 15, 16) and the road embeddings that our networks learn (Figure 18).
>
> **Scalability on Larger Graphs and More Complicated Queries**
>
> We already conduct experiments where we vary the scale of our model (Section B.2) and the density of points-of-interest in the road network (Section B.3). Our experiments also already study how our performance varies for simpler vs more complicated queries; for instance, compare the left and right-hand columns of Table 1.
>
> **Are there any potential limitations or challenges in applying the proposed method to real-world scenarios with more complex constraints and larger datasets?**
>
> Experiments are conducted on the US road network and on a dataset of natural language text queries with almost 1 billion unique sentences.

---

> > ### Comment · Reviewer_eXjd · 2023-11-22
> >
> > Thank you for addressing my questions. I do not have concerns about reproducibility. I increase my score of 'marginally above the acceptance threshold'.

---

### Official Review · Reviewer_vEqU · 2023-11-02

**Soundness:** 3 good
**Presentation:** 3 good
**Contribution:** 2 fair
**Rating:** 5
**Confidence:** 4

**Summary:**

The paper proposes using transformer models trained via next-vertex prediction for solving complex routing problems on graphs based on natural language queries.

Specifically, the paper formulates routing as an autoregressive next-vertex prediction task. Given a query, source, destination, and partial route, the goal is to predict the next vertex on the optimal route with transformer models. This transformer model jointly encodes the textual query and graph structure, and the paper trains the model on large amounts of routing data by decomposing optimal routes into next-vertex prediction examples. Beam search is used during inference to generate high-quality route candidates. Experiments on synthetic graphs as well as graphs from the OpenStreetMap repository prove the effectiveness of the proposed approach.

**Strengths:**

* Novel framing of routing as next-vertex prediction enables leveraging powerful transformer models and large-scale pretraining.

* Architecture jointly encodes textual queries and graph structure in an elegant way.

* Road embeddings allow scaling to massive real-world graphs.

* Training methodology based on next-vertex prediction is simple and efficient.

* Strong empirical results demonstrating high query fulfillment rates and near optimal routes.

**Weaknesses:**

* Relying only on local graph context could limit long-range reasoning.

The paper relies on representing each vertex and edge using features derived only from their local neighborhood in the graph. This local context allows the model to scale to massive graphs. However, it means the model may struggle with some types of long-range reasoning during routing. One example is to identify long shortcuts in the graph between distant vertices. With only local context, long shortcuts may not be recognized. Another example is reasoning about global properties of the graph such as determining a route which should avoid an entire region of the graph. In summary, the reliance on local context enables scaling to large graphs but inherently restricts the model's ability to reason about global graph structure and long-range dependencies. This could become a limitation for certain complex routing queries that require broader reasoning.

* Query fulfillment prioritized over efficiency may lead to suboptimal routes.

The paper prioritizes training the model to fulfill query constraints, even if that results in slightly less efficient routes. This makes sense for satisfying user requirements, but may produce routes that are longer or slower than necessary. In other words, the model lacks an explicit training signal to prefer efficient routing conditional on fulfilling the query. The lack of joint training on both query fulfillment and efficiency could lead to routes that satisfy the query but are not optimally efficient. This could become problematic in practice if it generates unnecessarily long routes over many queries. Explicitly optimizing for efficiency while still satisfying queries could improve the overall routing performance.

* Lack comparisons to other learning-based routing methods.

The paper does not provide direct comparisons to other machine learning approaches for routing, such as graph neural networks for routing, reinforcement learning for routing, attention-based models for vehicle routing. Without comparisons, it is hard to assess if the performance is truly state-of-the-art among learning-based methods.

**Questions:**

See the section of weaknesses above.

---

> ### Author Response · Authors · 2023-11-16
> **Response to Reviewer vEqU**
>
> We thank the reviewer for their response. We address your questions below.
>
> **Relying only on local graph context could limit long-range reasoning?**
>
> We found that local graph context is sufficient for real world routing problems. Our experiment in Section B.1 compares the performance of models allowed differing amounts of graph context, and finds that one does not gain any statistically significant amount of long-range reasoning capability by using a more than three-hop neighborhood as context.
>
> There are two main reasons local graph context is sufficient.
> 1) *Real-world road networks (which is what our experiments use) are easier to route on than worst-case graphs.* This is because travel time between locations usually approximates geographic distance up to small constant factors.
> 2) *In beam search, each candidate route allows our model to explore a different local region of the graph.* Our models may therefore be limited to using local information but can be strategic about where to collect this local information. In fact, we can think of beam search as performing some efficient graph search. Even with a 1-hop receptive field, a sufficiently large beam width lets one observe the entire graph before choosing a route. We also note that we train a secondary model to choose from the final set of candidate routes; this secondary model has the full context of each of the routes when scoring/ranking a solution.
>
> **Query fulfillment prioritized over efficiency may lead to suboptimal routes?**
>
> Our networks are neither directly trained to maximize efficiency nor directly trained to maximize query fulfillment. Rather, they are trained auto-regressively on ground-truth routes that reflect both efficient and fulfilling solutions. The models therefore already jointly optimize for both efficiency and fulfillment, weighting the two concerns in accordance with the training data. Moreover, all of our results (Tables 1, 2, 3) show minimal loss of travel time efficiency (<3%).
>
> **Lacking comparisons to attention-based, reinforcement learning, and graph neural network routing methods?**
>
> The main contribution of our paper is demonstrating that a next-vertex prediction approach yields surprisingly good results for natural language constrained routing. To the best of our knowledge, there are only two alternative approaches: the use of cost modifiers (learning edge costs so one can run an algorithm like Dijkstra’s) and the use of unguided search algorithms (e.g., using electrical flows). For the former, it is immediate that it is impossible for cost modifiers to handle point-of-interest-like queries, which would require route-level costs that cannot be decomposed additively into edge costs. For the latter, we do include an electrical flows baseline, which clearly demonstrates that the routing problems in our experiment are non-trivial and difficult. We are open to implementing other baseline alternatives to our next-vertex prediction approach, but are not aware of any others.
>
> We also want to note that there are ways that one could have implemented our next-vertex prediction approach differently. One could have used a different network to embed the receptive field, e.g. with a graph neural network rather than a transformer. However, we believe that we already have found a reasonable and effective implementation that demonstrates the efficacy of a next-vertex prediction approach; further architectural tunings are interesting but beyond the scope of this paper. Similarly, we agree that one could have used reinforcement learning to tune our model, in the same way that people finetune large language models with reinforcement learning; however, we see this slightly orthogonal question to be beyond the scope of this work.

---

### Author Response · Authors · 2023-11-21
**Note to reviewers**

We thank all the reviewers for their comments. Since the discussion period is ending in two days, we wanted to point to the individual responses we have included under each of your comments. Please let us know if you have any further questions and thanks again!

---

### Meta-Review · Area_Chair_V2ft · 2023-12-06

**Metareview:**

The paper addresses routing in a graph by using an LLM to generate a query, which in turn requires some representation of the (local) graph and the path. The performance is marginally higher than a rule-based system.

Reviewers generally find the idea contains some merits, but raise complains about the lack of proper baselines. Further, I agree with Reviewer P8sR that the paper fails to answer why LLM works for routing. Detailed analysis or probing test would make the paper interesting.

**Justification For Why Not Higher Score:**

The paper lacks proper baselines.
The paper appears shallow as it falls to show why LLM works for graph routing.

**Justification For Why Not Lower Score:**

N/A

---

### Decision · Program_Chairs · 2024-01-16

Reject